# Restoring Initial Noise Sensitivity in Text-to-Image Distillation via Geometric Alignment

**Huayang Huang**[1]  **Ruoyu Wang**[1]  **Jinhui Zhao**[1]  **Wei Deng**[2]
**Daiguo Zhou**[2]  **Jian Luan**[2]  **Yu Wu**[3][†]  **Ye Zhu**[4]

## Abstract

Generative distillation significantly accelerates text-to-image (T2I) generation by compressing multi-step trajectories into few-step student models while preserving perceptual quality. However, existing methods primarily optimize efficiency and output fidelity, often neglecting critical properties of the original trajectory. In this work, we identify a key missing property: sensitivity to initial noise, whose degradation impairs downstream control methods relying on noise-based optimization and manipulation. We trace this issue to standard distillation objectives that enforce pointwise output alignment, inadvertently flattening the input-output landscape and suppressing the teacher's local geometric structure. To address this, we propose Geometry-Aware Distillation (GAD), a sensitivity-preserving framework that aligns the local functional behavior of teacher and student models. Specifically, GAD matches Jacobian-vector products with respect to input noise, enabling the student to reproduce the teacher's differential response to perturbations. Extensive experiments across multiple T2I paradigms and noise-driven control tasks demonstrate that GAD significantly restores sensitivity and improves diversity while maintaining high visual fidelity. Code is available at https://github.com/Hannah1102/GAD.

## 1. Introduction

Iterative generative models, encompassing both Diffusion models (DMs) (Ho et al., 2020; Nichol et al., 2022) and

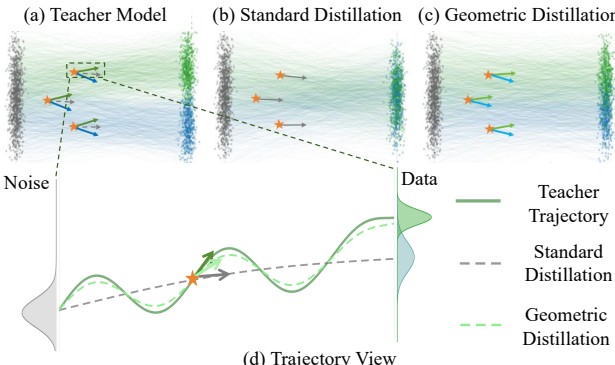

*Figure 1.* **Illustration of sensitivity degradation in diffusion distillation.** Top: While the Teacher (a) maps noise to distinct modes (green/blue clusters) with clear directional gradients (arrows), Standard Distillation (b) tends to average these modes, resulting in misaligned gradients. Our Geometry-Aware Distillation (c) successfully recovers the teacher's geometric structure. (d) Trajectory view: standard point-matching (grey dashed) learns a smoothed path with dampened gradients (flattened slope), whereas our GAD (green dashed) preserves the teacher's original curvature and sensitivity to initial noise.

Flow Matching (FM) (Lipman et al., 2023; Liu et al., 2023), have emerged as the dominant paradigm in modern text-to-image (T2I) synthesis. Despite their remarkable generation capabilities, the practical deployment of these models is often hindered by computationally expensive sampling procedures, which typically require tens to hundreds of Neural Function Evaluations (NFEs). To mitigate this latency, distillation techniques (Meng et al., 2023; Song et al., 2023; Sauer et al., 2024b) have been rapidly developed. By compressing complex multi-step trajectories into single or few-step mappings, these methods successfully reduce inference time by orders of magnitude, making real-time generation increasingly feasible.

Existing distillation methods (Lu et al., 2025; Chen et al., 2025) have largely focused on optimizing inference efficiency while maintaining perceptual fidelity. While successful under standard image quality metrics, this line of work implicitly treats the teacher model as a static input-output mapper and prioritizes alignment of final results. As a re-

[1]School of Computer Science, Wuhan University [2]MiLM Plus, Xiaomi Inc. [3]School of Artificial Intelligence, Wuhan University [4]Laboratoire d'Informatique (LIX), CNRS, École Polytechnique, IPP, France. Correspondence to: Yu Wu <wuyucs@whu.edu.cn>.

*Proceedings of the 43rd International Conference on Machine Learning*, Seoul, South Korea. PMLR 306, 2026. Copyright 2026 by the author(s).

sult, several intrinsic properties of the original generative process are often sacrificed, with recent studies highlighting degradation in generation diversity (Gandikota & Bau, 2025), negative prompt adherence (Nguyen et al., 2025), and trajectory invertibility (Starodubcev et al., 2024).

In this paper, we identify a critical yet underexplored limitation: *distilled models exhibit severely degraded sensitivity to the initial noise.* Rather than being a mere source of randomness, the initial noise acts as a structured prior that determines which specific trajectory the model follows within the vast manifold of images compatible with a given text prompt. Recent studies suggest that this noise-driven prior provides a complementary axis of control for generative aspects that are difficult to specify via text alone, such as precise spatial layouts (Ban et al., 2025), low-level visual attributes (Wang et al., 2025), and test-time quality enhancement (Eyring et al., 2024; Zhou et al., 2025d). More importantly, the creative potential of T2I models relies on the ability to produce diverse visual outputs from the same text prompt, a capability governed by the model's distinct responses to different noise initializations. However, we observe that compared to the teacher, where different random seeds induce diverse and controllable outputs, distilled models often produce highly correlated samples and respond weakly to noise perturbations. This loss of sensitivity not only hampers generative diversity (Gandikota & Bau, 2025; Cideron et al., 2025) but also undermines a broad class of noise-based control techniques (Xie et al., 2023; Zhou et al., 2025d; Ban et al., 2025), whose effectiveness relies on the model's precise responsiveness to input changes.

We attribute this phenomenon to the limitations of the standard distillation objectives. Specifically, most prior methods formulate distillation as pointwise output alignment, such as minimizing mean squared error (Luo et al., 2023; Sauer et al., 2024b) or reverse KL divergence (Yin et al., 2024b;a) between teacher and student predictions. While effective for matching average behavior, these objectives encourage the student to approximate a smoothed conditional expectation over potentially multi-modal outputs (Srinivas & Fleuret, 2018). As a result, the local geometric (Humayun et al., 2025) structure of the teacher mapping, particularly its differential response to input perturbations, is suppressed. As illustrated in Fig. 1, this averaging effect flattens the input-output landscape of the student (grey dashed line) and filter out the directional information encoded in the initial noise.

To address this limitation, we propose **Geometry-Aware Distillation (GAD)**, a sensitivity-preserving framework that aligns local functional behavior between teacher and student models. Instead of solely matching outputs, GAD enforces *response alignment* by constraining the student to replicate the teacher's Jacobian-vector products with respect to the input. This relational objective preserves local curvature and directional gradients, thus enabling small noise variations to induce meaningful and controllable changes in the synthesized visual outputs.

We conduct a rigorous validation of GAD across diverse generative architectures, spanning SD2 (UNet) (Rombach et al., 2022), PixArt-$\alpha$ (DiT) (Chen et al., 2024), and SANA (flow-based DiT) (Xie et al., 2025). We implement GAD within three representative distillation paradigms (output matching (Sauer et al., 2024a), distribution matching (Luo et al., 2025b) and score identity distillation (Zhou et al., 2025b)), benchmarking against 11 distilled baselines. Beyond standard metrics, we also employ several initial noise-based control tasks as a rigorous testbed for sensitivity recovery. Empirical results on layout control and training-free noise retrieval for alignment demonstrate that GAD significantly recovers the effectiveness of noise manipulation compared to standard distillation baselines. Furthermore, we show that restoring sensitivity naturally alleviates diversity degradation without compromising image fidelity, offering a unified solution that reconciles the trade-off between inference speed and generative controllability.

Our contributions are summarized as follows:

- We identify the initial noise sensitivity degradation in diffusion distillation and attribute it to the smoothing effect of standard pointwise alignment objectives.

- We propose Geometry-Aware Distillation (GAD), a model-agnostic framework that aligns the geometric structure via response alignment, effectively restoring the model's sensitivity to input noise.

- We extensively evaluate the controllability and diversity of distilled models, demonstrating that GAD achieves superior performance on multiple downstream tasks dependent on initial noise manipulation.

**Conflict of Interest Disclosure.** The authors declare that there are no conflicts of interest.

## 2. Background

### 2.1. Generative Models and Initial Noises

Existing diffusion (Ho et al., 2020) and Flow Matching models (Lipman et al., 2023) can be viewed as learning a velocity field $v_\phi$ that defines a probability flow ODE (PF-ODE) (Song et al., 2021). Let $\mathbf{x}_t$ denote the state at time $t \in [0, 1]$, where $t = 1$ corresponds to the prior distribution $\mathcal{N}(\mathbf{0}, \mathbf{I})$ (pure noise) and $t = 0$ corresponds to the real data $\mathbf{x}_0 \sim p_{\text{data}}$ (clean images). The generative process is governed by the ODE:

$$\frac{d\mathbf{x}_t}{dt} = v_\phi(\mathbf{x}_t, t), \quad \mathbf{x}_1 \sim \mathcal{N}(\mathbf{0}, \mathbf{I}). \tag{1}$$

In standard diffusion models, $v_\phi$ is derived from the predicted score function $\epsilon_\phi(\mathbf{x}_t, t)$, whereas in Flow Matching, $v_\phi$ is learned directly by regressing target vector fields. We denote the exact solution of Eq. 1 from time $t = 1$ to $t = 0$ as the **Teacher Mapping** $\Phi_T : \mathbb{R}^d \to \mathbb{R}^d$. In practice, sampling from $\Phi_T(\mathbf{x}_1)$ requires $N \in [20, 100]$ function evaluations (NFEs), which creates the latency bottleneck.

Beyond serving as a stochastic starting point, the initial noise $\mathbf{x}_1$ is fundamental to diffusion models. Under the teacher mapping $\Phi_T$, different initializations induce distinct PF-ODE trajectories, leading to diverse samples even under identical conditioning (Sadat et al., 2024; Um & Ye, 2025). More importantly, the expressive dependence of $\Phi_T$ on $\mathbf{x}_1$ enables a wide range of noise-based control mechanisms, such as diversity modulation (Um et al., 2025), spatial and semantic steering (Xie et al., 2023; Li et al., 2025), and retrieval or manipulation of initial noises (Wang et al., 2025; Zhou et al., 2025d) for targeted generation. This sensitivity to initial noise is a key factor underlying the flexibility of modern diffusion models, as shown in Fig. 2 (c).

## 2.2. Diffusion Distillation

Diffusion distillation aims to learn a **Student Model** $\Phi_S(\cdot; \theta) : \mathbb{R}^d \to \mathbb{R}^d$, parameterized by $\theta$, that approximates the teacher's trajectory $\Phi_T$ in a single or few steps. Given an input noise $\mathbf{z} \sim \mathcal{N}(\mathbf{0}, \mathbf{I})$, the same as the initial state $\mathbf{x}_1$ in the PF-ODE formulation, the student generates a sample $\hat{\mathbf{x}} = \Phi_S(\mathbf{z}; \theta)$. While specific algorithms differ, most existing distillation methods (Kim et al., 2024; Luo et al., 2025a; Lu et al., 2025) can be generalized as minimizing a divergence or distance metric $\mathcal{D}$ between the student's mapping $\Phi_S$ and a target signal $\Phi_T$ derived from the teacher:

$$\mathcal{L}_{\text{base}} = \mathbb{E}_{\mathbf{z} \sim p(\mathbf{z})} \big[ \mathcal{D}\big(\Phi_S(\mathbf{z}), \Phi_T(\mathbf{z})\big) \big], \quad (2)$$

where $\Phi_T(\mathbf{z})$ represents the supervision signal from the teacher (e.g., the ODE trajectory endpoint or a score estimate). Depending on the specific paradigm, $\mathcal{L}_{\text{base}}$ takes various forms: *Output Matching* (Lin et al., 2024; Song & Dhariwal, 2024) minimizes the discrepancy between student output and teacher trajectory points at certain timesteps, using a regression-based loss (e.g., $L_2$ distance or LPIPS loss). *Adversarial Alignment* (Sauer et al., 2024a;b) employs a discriminator to enforce distributional alignment when paired data is scarce or to enhance perceptual quality. *Score Matching* (Yin et al., 2024b;a) uses the teacher's score function to provide gradients for the student, avoiding explicit trajectory generation. Details about the baseline methods used in our experiments are provided in Appendix C.

## 3. Geometric Gap in Distillation

Standard distillation objectives (Eq. 2) focus on *pointwise alignment*, ensuring that the student $\Phi_S(\mathbf{z})$ matches the

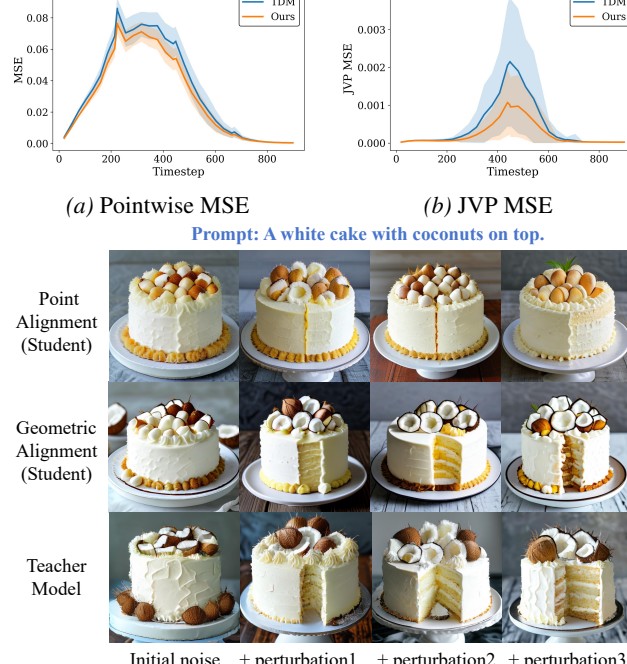

*(a)* Pointwise MSE      *(b)* JVP MSE

**Prompt: A white cake with coconuts on top.**

Point Alignment (Student)

Geometric Alignment (Student)

Teacher Model

Initial noise   + perturbation1   + perturbation2   + perturbation3

*(c)* Visual sensitivity to identical initial-noise perturbations.

*Figure 2.* **Geometric gap in distillation.** Comparison between baseline TDM (Blue) and our method (Orange). While the baseline achieves comparable pointwise MSE to our method **(a)**, it suffers more from high geometric error **(b)** and attenuated variations to input perturbations **(c)**.

teacher $\Phi_T(\mathbf{z})$ for individual inputs. However, this objective treats inputs $\mathbf{z}$ independently and does not explicitly constrain the *functional landscape* between samples or the local geometry around them. This mirrors a known challenge in classical knowledge distillation, where capturing the relational knowledge between samples is often more crucial than mimicking absolute output values (Park et al., 2019; Tung & Mori, 2019). In the generative context, we hypothesize that this independent treatment overlooks the differential structure of the generative mapping, potentially leading to a misalignment of local gradients. As a consequence, while the student may produce high-fidelity average outputs, it fails to replicate how the teacher responds to input variations-a phenomenon we define as a collapse in *initial noise sensitivity*.

**Diagnostic Experiment.** To empirically verify this hypothesis, we conduct a pilot analysis comparing a standard TDM (Luo et al., 2025b) baseline against our GAD (with additional geometric alignment objective). We track two metrics after distillation: *1) Point Alignment:* Measured by Mean Squared Error (MSE) between $\Phi_S(\mathbf{z})$ and $\Phi_T(\mathbf{z})$. *2) Geometric Alignment:* Measured by the error in Jacobian-Vector Products (JVP), which represents the mismatch in response to input perturbations. The JVP MSE is computed following

*Table 1.* **Direct measurements of geometric alignment.** Metrics are computed on PixArt-$\alpha$ with baseline TDM.

| Method | JVP Cos.↑ | Jac. Norm↑ | Spec. KL↓ | JVP MSE↓ |
|---|---|---|---|---|
| Teacher | 1.000 | 1.000 | 0.000 | 0.000 |
| TDM | 0.012 | 0.98 | 0.008 | 0.0003 |
| Ours | **0.014** | **0.99** | **0.006** | **0.0002** |

the directional derivative objective in Eq. 6. Experiments are performed on PixArt-$\alpha$ (Chen et al., 2024) using 128 COCO prompts (Caesar et al., 2018).

**Observation.** As shown in Fig. 2(a), both methods achieve low MSE, indicating that standard distillation is effective at aligning outputs in a *global expectations* sense. Notably, our geometric constraint does not compromise and even marginally improves point alignment. However, a discrepancy appears in Fig. 2 (b) (JVP Error). The baseline exhibits high error in replicating the teacher's gradient response. It learns to map $\mathbf{z}$ to the correct image, but via a "smoother" functional path that lacks the teacher's crisp sensitivity. This manifests visually in Fig. 2(c), where the point-aligned student exhibits noticeably dampened and less structured changes compared to the teacher. In contrast, the student trained with our geometric alignment responds in a manner closely matching the teacher, indicating that the local input-output geometry is effectively preserved.

**Direct Measurement of Geometric Alignment.** To provide a more direct assessment of the geometric alignment between teacher and student models, we complement the JVP MSE analysis with additional metrics: JVP Cosine Similarity, Jacobian Norm Ratio, and Spectral KL divergence. These metrics quantify the local differential response and spectral properties of the student relative to the teacher. As summarized in Tab. 1, incorporating GAD consistently improves alignment across all measures, demonstrating that the student closely replicates the teacher's local geometry.

**Conclusion.** These empirical findings reveal that high-fidelity generation (low MSE) does not imply correct underlying dynamics. To bridge this gap, we propose Geometry-Aware Distillation (GAD), which we detail in the following.

## 4. Method

In this section, we introduce Geometry-Aware Distillation (GAD), which serves as a model-agnostic regularization term that can be seamlessly integrated with existing distillation paradigms. We first present the general theoretical formulation of GAD centered on functional geometry alignment, followed by its specific instantiations within various established distillation frameworks.

### 4.1. Geometry-Aware Distillation (GAD)

To restore sensitivity, we propose to explicitly align the local functional behavior of the student $\Phi_S$ with that of the teacher $\Phi_T$. Ideally, we seek to minimize the squared Frobenius norm, which enforces element-wise alignment between the Jacobian matrices:

$$\mathcal{L}_{\text{Jacobian}} = \mathbb{E}_{\mathbf{z} \sim p(\mathbf{z})} \left[ \|\mathbf{J}_{\Phi_S}(\mathbf{z}) - \mathbf{J}_{\Phi_T}(\mathbf{z})\|_F^2 \right], \quad (3)$$

where $\mathbf{J}_{\Phi}(\mathbf{z}) = \nabla_{\mathbf{z}}\Phi(\mathbf{z})$. Following the general formulation in Eq. 2, we let $\Phi(\mathbf{z})$ denote the model's mapping function, which can be instantiated as the predicted trajectory endpoint $\hat{\mathbf{x}}_0$ or the score estimate $\epsilon$ depending on the specific distillation setting. The input $\mathbf{z}$ is interpreted broadly: it corresponds to the initial Gaussian noise for one-step models, and the intermediate noisy latent state $\mathbf{x}_t$ for few-step distillation. Thus, GAD enforces geometric consistency across the entire sampling trajectory. However, computing the full Jacobian matrix for high-dimensional image data ($d \approx 10^5$) is computationally prohibitive.

**Jacobian-Vector Product (JVP) Alignment.** Instead of materializing the full matrix, we propose to align directional derivatives, which capture the essential local geometry. For a random perturbation vector $\mathbf{v} \sim \mathcal{N}(\mathbf{0}, \mathbf{I})$, the Jacobian-vector product (JVP) $\mathbf{J}_{\Phi}(\mathbf{z}) \cdot \mathbf{v}$ (Pearlmutter, 1994) characterizes the model's response to the perturbation. We therefore formulate our Geometry-Aware Distillation (GAD) loss as matching these directional responses:

$$\mathcal{L}_{\text{GAD}}(\theta) = \mathbb{E}_{\mathbf{z},\mathbf{v}} \left[ \|\nabla_{\mathbf{z}}\Phi_S(\mathbf{z};\theta) \cdot \mathbf{v} - \nabla_{\mathbf{z}}\Phi_T(\mathbf{z}) \cdot \mathbf{v}\|_2^2 \right]. \quad (4)$$

Theoretically, matching the response to random perturbations $\mathbf{v} \sim \mathcal{N}(\mathbf{0}, \mathbf{I})$ is equivalent to minimizing the Frobenius norm of the Jacobian difference in expectation (Hutchinson, 1989; Czarnecki et al., 2017). Thus, this objective implicitly aligns the full Jacobian geometry.

**Efficient Approximation via Finite Differences.** While exact JVP can be computed via forward-mode automatic differentiation (Baydin et al., 2018; Finlay et al., 2020), it can be memory-intensive or incompatible with certain black-box teacher implementations. To ensure broad applicability and efficiency, we approximate the JVP using a finite difference scheme, as illustrated in Fig. 3. We perturb the input $\mathbf{z}$ by a small magnitude $h$ in the direction of $\mathbf{v}$:

$$\nabla_{\mathbf{z}}\Phi(\mathbf{z}) \cdot \mathbf{v} \approx \frac{\Phi(\mathbf{z} + h\mathbf{v}) - \Phi(\mathbf{z})}{h}, \quad (5)$$

Substituting this into Eq. 4 and absorbing the constant scaling factor $1/\epsilon^2$ into the loss weight hyperparameter, we

**(a) Existing Diffusion Distillation (Pointwise Alignment)**

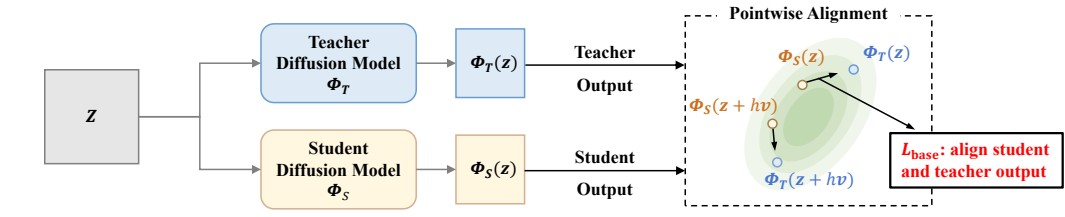

**(b) Ours Method (Pointwise + Geometric Alignment)**

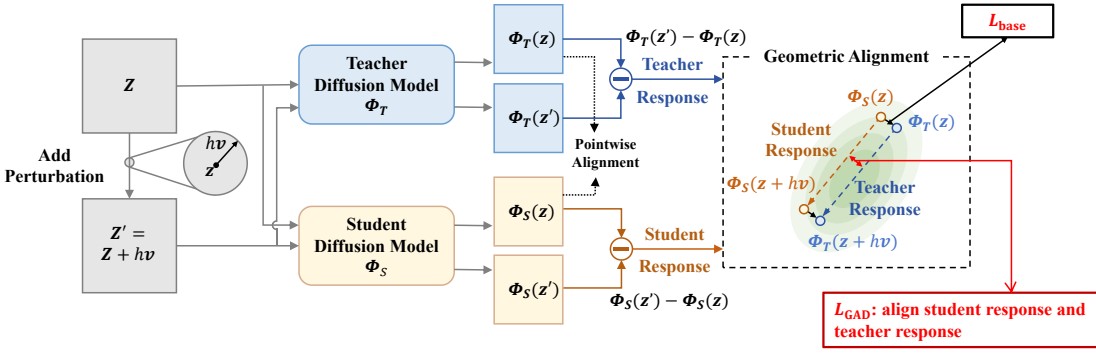

*Figure 3.* **Overview of Geometry-Aware Distillation (GAD).** (a) Existing distillation paradigms typically focus on individual pointwise alignment, which often leads the student to learn an "averaged" direction between $\Phi_T(\mathbf{z})$ and $\Phi_T(\mathbf{z}')$, thus resulting in a flattened response and loss of diversity. (b) Our GAD complements the standard loss (dashed) by aligning paired inputs $(\mathbf{z}, \mathbf{z}')$ to align the *Response Vectors*. By explicitly matching the relative change $(\Phi(\mathbf{z}) - \Phi(\mathbf{z}'))$, GAD ensures the teacher's output variation is inherited by the student.

arrive at our practical objective:

$$\mathcal{L}_{\text{GAD}}(\theta) = \mathbb{E}_{\mathbf{z},\mathbf{v}}\Big[\big\|\underbrace{\big(\Phi_S(\mathbf{z}') - \Phi_S(\mathbf{z})\big)}_{\text{Student Response}}$$
$$- \underbrace{\text{sg}\big(\Phi_T(\mathbf{z}') - \Phi_T(\mathbf{z})\big)}_{\text{Teacher Response}}\big\|_2^2\Big], \quad (6)$$

where $\mathbf{z}' = \mathbf{z} + h\mathbf{v}$, and $\text{sg}(\cdot)$ denotes the stop-gradient operator to fix the teacher's trajectory. Unlike standard distillation (Fig. 3(a)), which processes a single input to align pointwise outputs, our method (Fig. 3(b)) processes a paired input: the original noise $\mathbf{z}$ and a perturbed version $\mathbf{z}'$. Both inputs are fed into the teacher and student models. We then compute the *response vectors* by taking the difference between the paired outputs. The GAD objective strictly enforces that the Student Response $(\Phi_S(\mathbf{z}) - \Phi_S(\mathbf{z}'))$ matches the Teacher Response $(\Phi_T(\mathbf{z}) - \Phi_T(\mathbf{z}'))$ This loss forces the student to replicate the *relative change* of the teacher's output given a shift in input, effectively locking the student's local curvature to the teacher's manifold.

### 4.2. Unified Training Framework

A distinct advantage of our formulation is that $\mathcal{L}_{\text{GAD}}$ is independent to the pointwise alignment objective. This allows us to apply GAD as a plug-and-play regularizer across diverse distillation paradigms. The total training objective is:

$$\mathcal{L}_{\text{total}} = \mathcal{L}_{\text{base}} + \lambda\mathcal{L}_{\text{GAD}}, \quad (7)$$

where $\lambda$ is a balancing hyperparameter. We instantiate this framework across two representative settings:

**Output Matching:** In methods like ADD (Lin et al., 2024) or LADD (Sauer et al., 2024a), the student $\Phi_S$ directly predicts the trajectory endpoint. Let $\hat{\mathbf{x}}_0 = f_\theta(\mathbf{x}_t, t, c)$ be the model prediction, in which $t$ is the timestep and $c$ is the text condition. The GAD loss ensures that the change in the predicted image mirrors the teacher's response to a perturbed latent $\mathbf{x}'_t = \mathbf{x}_t + h\mathbf{v}$:

$$\mathcal{L}_{\text{GAD}}^{\text{out}} = \mathbb{E}_{\mathbf{x}_t,\mathbf{v},t,c}\Big[\big\|\Delta\hat{\mathbf{x}}_0^S(\mathbf{x}_t,\mathbf{v}) - \text{sg}\big(\Delta\hat{\mathbf{x}}_0^T(\mathbf{x}_t,\mathbf{v})\big)\big\|_2^2\Big],$$
$$(8)$$

where $\Delta\hat{\mathbf{x}}_0(\mathbf{x}_t,\mathbf{v}) = f(\mathbf{x}_t + h\mathbf{v}, t, c) - f(\mathbf{x}_t, t, c)$.

**Score-based Alignment:** For paradigms that align distributions by matching score fields across all timesteps, such as DMD (Yin et al., 2024b) and SiD (Zhou et al., 2025a;c), the base objective $\mathcal{L}_{\text{base}}$ typically minimizes a divergence between the student-generated distribution $p_{\text{fake}}$ and the real data distribution $p_{\text{real}}$. DMD achieves this by minimizing the KL divergence, while SiD employs the Fisher divergence, enabling a data-free distillation process. In this context, the gradient of the loss involves two score estimators: $\epsilon_{\text{real}}$, the pre-trained teacher model, and $\epsilon_{\text{fake}}$, an auxiliary score

estimator trained to predict the noise in samples generated by the student. GAD introduces a higher-order geometric constraint by matching the directional variation of these score fields. The GAD gradient is formulated as:

$$\nabla_\theta \mathcal{L}_{\text{GAD}}^{\text{score}} = \mathbb{E}_{\mathbf{x}_t, \mathbf{v}, t, c} \left[ \Delta\epsilon_{\text{fake}}(\mathbf{x}_t, \mathbf{v}) - \Delta\epsilon_{\text{real}}(\mathbf{x}_t, \mathbf{v}) \right] \frac{\partial \mathbf{x}_t}{\partial \theta},$$
$$(9)$$

where the Score Response $\Delta\epsilon$ is the difference between noise predictions at the original and perturbed locations:

$$\Delta\epsilon(\mathbf{x}_t, \mathbf{v}) = \epsilon(\mathbf{x}_t + h\mathbf{v}, t, c) - \epsilon(\mathbf{x}_t, t, c). \quad (10)$$

Intuitively, while $\mathcal{L}_{\text{base}}$ aligns the first-order moments (ensuring the student moves toward high-density regions of the teacher's distribution), $\mathcal{L}_{\text{GAD}}^{\text{score}}$ aligns the local curvature and divergence of the score fields. This ensures that the geometric structure of the synthesized manifold consistently follows the teacher's guidance even under local perturbations. The comprehensive list of hyperparameters and model-specific configurations is included in Appendix D.

## 5. Experiment

We first analyze the noise sensitivity of distilled models, followed by an evaluation of the impact of GAD on general generation quality. Finally, we show comparisons on downstream tasks that rely on noise sensitivity for control.

### 5.1. Loss of Sensitivity in Distillation

We first design a diagnostic experiment to empirically verify our core hypothesis that existing diffusion distillation methods suffer from a collapse in *initial noise sensitivity*: variations in the initial noise no longer induce sufficiently distinguishable changes in the generated images.

**Experimental Protocol.** We formulate sensitivity as a *seed identifiability* problem by asking: can a model distinguish between images generated from different initial seeds? We generate images using 10 fixed seeds with MS-COCO (Lin et al., 2014) prompts and train an EfficientFormer-L3 (Li et al., 2022) classifier to predict the source seed. For each seed, we generate 500 training, 100 validation, and 100 testing images. We report two metrics: (1) *Self-Identifiability*: the classifier is trained and tested on the same model to measure the intrinsic noise sensitivity; and (2) *Teacher Alignment*: the classifier is trained on the Teacher (SD2) and tested on the student models, probing whether the distilled model preserves the semantic trajectory of the teacher.

**Results.** As illustrated in Tab. 2, the Multi-step Teacher exhibits high classification accuracy, indicating that the teacher preserves strong and discriminative variations across different initial seeds. In contrast, classifiers trained and tested on standard distilled models exhibit significantly lower accuracy. Our model, which is built upon the LADD with

*Table 2.* **Quantitative analysis of seed sensitivity and teacher alignment.** Classification accuracy (%) of predicting initial seeds from generated images. Self-Identifiability: train/test on self. Teacher Alignment: train on teacher, test on student.

| Distilled Model | Self-Identifiability | Teacher Alignment |
|---|---|---|
| SD2 (Teacher) | 93.70% | - |
| SD-Turbo (Sauer et al., 2024b) | 77.80% | 63.20% |
| SwiftBrush (Nguyen & Tran, 2024) | 52.90% | 45.80% |
| SwiftBrushv2 (Dao et al., 2024) | 76.50% | 67.60% |
| TCD (Zheng et al., 2024) | 87.30% | 84.50% |
| LADD (Sauer et al., 2024a) | 87.60% | 83.70% |
| **Ours** | **92.40%** | **87.40%** |

*Table 3.* **Impact on general generation quality.** Our GAD regularization even slightly improves generation quality.

| Method | Architecture | Steps | CLIP ↑ | Pickscore ↑ |
|---|---|---|---|---|
| *Setting A: Output Matching with Adversarial Alignment on UNet* | | | | |
| Teacher (SD v2.1) | UNet | 50 | 34.56 | 21.7022 |
| LADD (Sauer et al., 2024a) | UNet | 1 | 32.58 | 20.8515 |
| **LADD + Ours** | UNet | 1 | **32.68** | **21.3790** |
| *Setting B: Distribution Matching on DiT* | | | | |
| Teacher (PixArt-$\alpha$) | DiT | 20 | 33.41 | 22.2922 |
| TDM (Luo et al., 2025b) | DiT | 4 | 33.43 | 22.1007 |
| **TDM + Ours** | DiT | 4 | **33.52** | **22.2704** |
| *Setting C: Score Identity Distillation on Flow-DiT* | | | | |
| Teacher (SANA) | Flow-DiT | 20 | 35.16 | 22.4056 |
| SiD (Zhou et al., 2025b) | Flow-DiT | 4 | 32.75 | 21.7629 |
| **SiD + Ours** | Flow-DiT | 4 | **34.40** | **22.0735** |

additional geometric sensitivity regularization, restores this sensitivity to 92.40%, closely matching the teacher. In terms of Teacher Alignment, our method achieves an accuracy of 87.40%, outperforming baselines. This result demonstrates that our GAD not only recovers the magnitude of noise sensitivity but also ensures that the *directional* response to noise perturbations remains consistent with the teacher model.

### 5.2. Impact on General Generation Quality

To ensure geometric regularization does not compromise image fidelity, we evaluate PickScore (Kirstain et al., 2023) and CLIP Score (Radford et al., 2021) on 1000 MS-COCO prompts (Lin et al., 2014) to assess image fidelity and text-image alignment. As reported in Tab. 3, our GAD slightly improves general generation quality across all evaluated settings. For instance, adding GAD to SiD (Zhou et al., 2025b) increases the CLIP Score from 32.75 to 34.40. This consistency proves that our method effectively restores noise sensitivity without disrupting the generative capabilities established by the base distillation methods. Unlike standalone methods that might be tied to specific architectures, GAD acts as a versatile enhancer that can be seamlessly integrated into existing state-of-the-art pipelines.

**Analysis.** We attribute this enhancement to GAD's better generalization. Unlike standard pointwise distillation that treats inputs independently and risks overfitting to isolated samples, GAD enforces local neighborhood consistency.

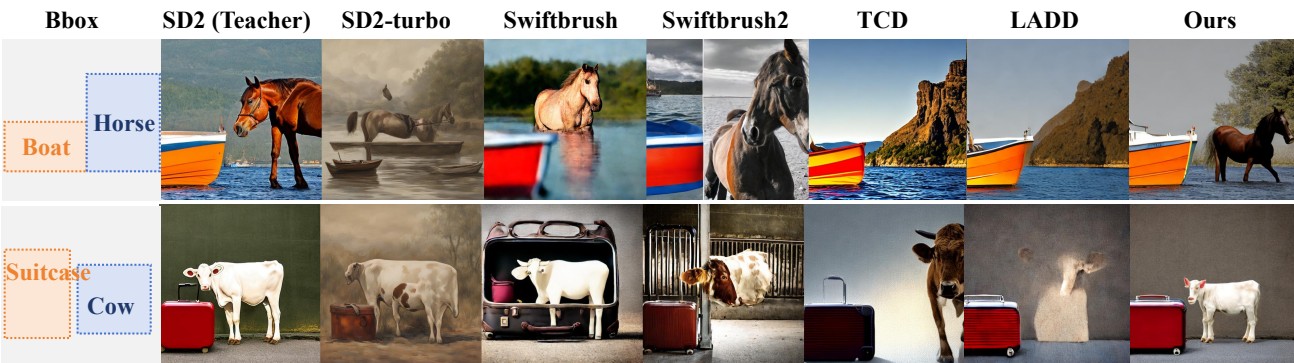

*Figure 4.* **Qualitative comparison of layout control.** The left column shows the target bounding boxes. The text prompts are "A horse and a boat." (first row) and "A cow and a suitcase." (second row).

*Table 4.* **Average cumulative trajectory deviation from the teacher on PixArt-$\alpha$.** Lower values indicate that the student's trajectory is more consistent with the teacher.

| Method | Stage 1 $(t = 0.75)\downarrow$ | Stage 2 $(t = 0.5)\downarrow$ | Stage 3 $(t = 0.25)\downarrow$ | Final $(t = 0)\downarrow$ |
|---|---|---|---|---|
| TDM | 0.016 | 0.216 | 0.433 | 0.491 |
| TDM + Ours | **0.014** | **0.184** | **0.373** | **0.427** |

This yields a smoother, more faithful approximation of the teacher's manifold. Consequently, GAD produces denoising trajectories that are more consistent with the teacher on unseen inputs. As empirically validated in Tab. 4, tracking the discrepancy between denoised latents on 200 unseen prompts reveals that GAD consistently reduces cumulative trajectory deviation, achieving a 13% lower final error. This tighter trajectory alignment preserves teacher-consistent dynamics, explaining the observed fidelity gains.

### 5.3. Downstream task 1: Layout Control

We next quantitatively evaluate whether restoring noise sensitivity translates into improved controllability. We consider a training-free layout control task (Xie et al., 2023), where the attention-based spatial constraints are injected exclusively through the initial noise.

**Experimental Protocol.** We evaluate performance using 800 prompts from COCO dataset (Caesar et al., 2018) with bounding box annotations from (Xie et al., 2023). Layout fidelity is measured via Average Precision (AP) and $AP_{50}$ (Li et al., 2021) using YOLOv4 (Bochkovskiy et al., 2020), alongside CLIP score (Radford et al., 2021) for semantic alignment. We implement our method on top of the LADD framework (Sauer et al., 2024a) and compare it against representative distillation baselines.

**Results.** Tab. 5 summarizes the results. The teacher model (SD v2.1) (Rombach et al., 2022) establishes an upper bound with an AP of 6.6. Most distilled baselines exhibit a substantial performance drop (e.g., SD-Turbo drops to 3.0 AP),

*Table 5.* **Quantitative comparison on layout control.** We report AP to measure layout fidelity, and CLIP score for text alignment.

| Model | AP $\uparrow$ | $AP_{50}$ $\uparrow$ | CLIP $\uparrow$ |
|---|---|---|---|
| SD2 (Teacher Model) | 6.6 | 21.3 | 0.3333 |
| SD-turbo (Sauer et al., 2024b) | 3.0 | 8.4 | **0.3237** |
| Swiftbrush (Nguyen & Tran, 2024) | 3.8 | 12.6 | 0.3147 |
| Swiftbrushv2 (Dao et al., 2024) | 2.9 | 10.6 | 0.3203 |
| TCD (Zheng et al., 2024) | 4.8 | 18.0 | 0.3169 |
| LADD (Sauer et al., 2024a) | 5.0 | 17.4 | 0.3187 |
| **Ours** | **5.8** | **20.6** | 0.3184 |

indicating that spatial information in the initialization is largely washed out. In contrast, our method achieves the strongest performance among students, significantly outperforming the base LADD model and recovering 87% of the Teacher's layout accuracy (AP 5.8). Qualitative results in Fig. 4 further confirm that while baselines often suffer from object neglect or spatial entanglement, our method faithfully respects the prescribed bounding boxes. These results demonstrate that preserving geometry effectively prevents the loss of spatial constraints encoded in the initial noise.

### 5.4. Downstream task 2: Generation Diversity

A common failure mode of distilled diffusion models is conditional mode collapse (Yin et al., 2024b), where different random seeds yield nearly identical outputs for a fixed prompt. We evaluate whether restoring noise sensitivity alleviates this issue.

**Experimental Protocol.** Following (Kirchhof et al., 2025), we use the CC12M dataset (Changpinyo et al., 2021). For each text prompt, we generate 8 images using different random seeds. We employ Vendi Score (Dan Friedman & Dieng, 2023) to measure diversity and CLIP score for text-image alignment. We incorporate various publicly available distilled models for a comprehensive comparison. For SD v2.1, we evaluate SD-Turbo, SwiftBrush/v2, TCD, and LADD. For PixArt-$\alpha$, we incorporate DMD (Yin et al.,

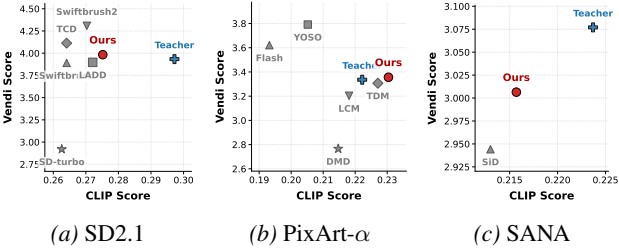

*(a)* SD2.1     *(b)* PixArt-$\alpha$     *(c)* SANA

*Figure 5.* **Diversity vs. fidelity trade-off.** Vendi Score (Diversity) vs. CLIP Score across three architectures. Baseline methods (grey) exhibit a clear trade-off, whereas our method (red) consistently lies in the upper-right region close to the Teacher (blue).

2024b), LCM (Luo et al., 2023), YOSO (Luo et al., 2025a), FLASH (Chadebec et al., 2025), and TDM (Luo et al., 2025b). For SANA, we utilize the SiD (Zhou et al., 2025b) method. Our method is implemented on top of LADD, TDM and SiD for SD2, PixArt-$\alpha$, and SANA.

**Results.** Fig. 5 illustrates the trade-off landscape. Baseline methods often struggle to balance these metrics, either suffering from reduced Vendi Scores or sacrificing CLIP alignment to sustain diversity. In contrast, GAD effectively mitigates this trade-off, consistently residing in the favorable upper-right region near the Teacher. Qualitative inspection (Fig. 6 (a)) confirms that GAD produces richer semantic variations, validating that restoring noise sensitivity recovers the teacher's semantic exploration capabilities.

### 5.5. Downstream task 3: Transfer of Controllability

We further evaluate the functional consistency with the teacher via NoiseQuery (Wang et al., 2025). This method retrieves the optimal initial noise that best matches the target attributes from a pre-computed database.

**Experimental Protocol.** We construct a noise-feature database using the teacher model by sampling $10^4$ noise vectors and storing their generated features. At inference time, we retrieve the optimal noise $\mathbf{z}^*$ based on the teacher feature, and directly apply the retrieved noise to the student model *without any adaptation*. This establishes a challenging zero-shot transfer setting, where successful control requires the student to share the same noise-to-image geometry as the teacher. The experiments are conducted on DrawBench (Saharia et al., 2022) dataset. We report CLIP Score, HPSv2 (Wu et al., 2023), and PickScore (Kirstain et al., 2023) to measure the alignment and aesthetic quality.

**Results.** Quantitative results under semantic retrieval are summarized in Tab. 6. It shows that standard distilled models often fail to benefit from the teacher's optimized noise. In contrast, applying our method improves performance in most cases. This confirms that GAD not only restores sensitivity but effectively aligns the student's functional

*Table 6.* **Quantitative results on zero-shot controlling.** Evaluation of control transferability by applying the teacher's optimal noise retrieved by NoiseQuery to the student during inference.

| Model | CLIPScore ↑ | HPSv2 ↑ | PickScore ↑ |
|---|---|---|---|
| *SD2 (UNet)* | | | |
| Teacher (Rombach et al., 2022) | 31.62 | 0.258 | 21.594 |
| LADD (Sauer et al., 2024a) | 30.79 | 0.223 | 20.839 |
| LADD+Ours | **30.87** | **0.232** | **20.884** |
| *PixArt-$\alpha$ (DiT)* | | | |
| Teacher (Chen et al., 2024) | 30.97 | 0.280 | 22.174 |
| TDM (Luo et al., 2025b) | 30.51 | 0.271 | 21.900 |
| TDM+Ours | **30.57** | **0.274** | **21.980** |
| *SANA (Flow-based DiT)* | | | |
| Teacher (Xie et al., 2025) | 32.33 | 0.289 | 22.523 |
| SiD (Zhou et al., 2025b) | **31.61** | 0.249 | 22.081 |
| SiD+Ours | 31.59 | **0.289** | **22.190** |

*Table 7.* **Ablation study on Geometry-Aware Distillation (GAD).** Metrics are reported on PixArt-$\alpha$ with baseline distillation TDM.

| Method Setting | Detail | Vendi ↑ | CLIP ↑ | PickScore ↑ |
|---|---|---|---|---|
| **Baseline (TDM)** | - | 2.7187 | 33.42 | 22.1240 |
| **+ Noise Augmentation** | No constraint | 2.6355 | 33.41 | 22.1924 |
| **+ Diversity Reg.** | Blindly Active | **3.0559** | 31.76 | 21.3556 |
| **+ Ours** | Geometric Alignment | 2.7967 | **33.53** | **22.2741** |

landscape with the teacher to enable zero-shot transfer of test-time enhancement. Fig. 6(b) visualizes retrieval for low-level attributes (e.g., blue hue, high brightness). The leftmost column visualizes the unconditional image generated from $\mathbf{z}^*$, revealing the intrinsic low-level bias encoded in the noise. Using a minimally informative prompt (e.g., "*A car*"), our method more faithfully preserves the low-level attributes specified by the retrieved noise.

### 5.6. Ablation Study

**Strategy Variations.** To validate the effectiveness of GAD, we compare against two alternative baselines on PixArt-$\alpha$ with 1k COCO prompts. **(1) Noise Augmentation:** We train the student with perturbed inputs $\mathbf{z}' = \mathbf{z} + \epsilon\mathbf{v}$ but without the Jacobian matching loss. As shown in Tab. 7, this setting actually leads to a decrease in Vendi Score. This suggests that without an explicit target response, the student learns to be *invariant* to input noise rather than sensitive to it. **(2) Diversity Regularization:** We apply a blind repulsion loss (maximizing $\|\Phi_S(z') - \Phi_S(z)\|$) to force sensitivity. While this yields the highest Vendi Score, it comes at a cost to semantic alignment (CLIP Score drop). This confirms that blindly forcing variance destroys the precise mapping required for high-fidelity generation. In contrast, GAD acts as a guided sensitivity restoration, where the direction of change is dictated by the teacher's Jacobian, ensuring that diversity is semantically meaningful. In addition, a detailed sensitivity analysis regarding the perturbation scale $h$ and the weight parameter $\lambda$ can be found in Appendix A.

**Supervised Timesteps.** To understand the temporal sen-

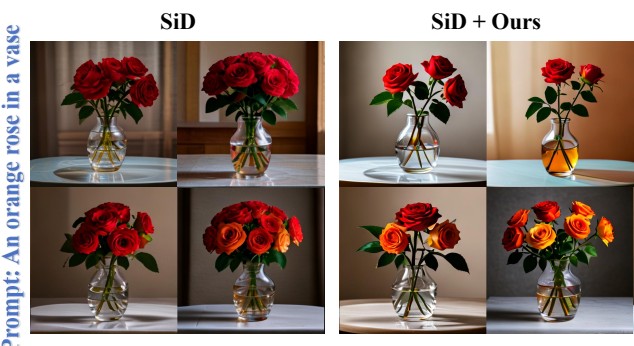
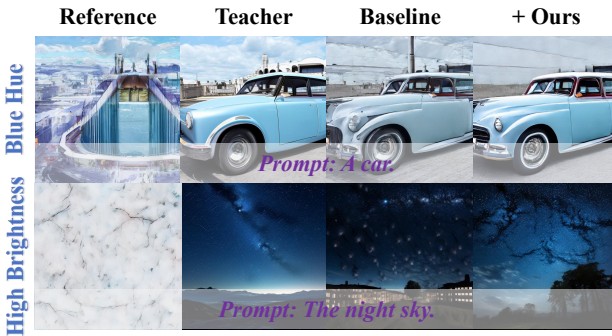

(a) Visualization of generation diversity        (b) Low-level control via NoiseQuery

*Figure 6.* **Visualization of diversity and low-level control.** (a) Generated images of baseline distilled models (SiD) (Zhou et al., 2025b) and ours under the same set of initial noises. (b) Zero-shot control via NoiseQuery (Wang et al., 2025): retrieving noise for "Blue Hue" and "High Brightness" from the teacher.

*Table 8.* **Ablation on GAD timesteps.** The performance of applying GAD to different noise intervals (sparse supervision) on PixArt-$\alpha$ with baseline TDM.

| GAD Timesteps | CLIP ↑ | PickScore ↑ | Vendi ↑ |
|---|---|---|---|
| Baseline (No GAD) | 33.43 | 22.1007 | 2.5967 |
| High noise ($t > 600$) | **33.59** | 22.2385 | 2.6686 |
| Mid noise ($t \in [200, 600]$) | 33.45 | 22.1274 | 2.7140 |
| Low noise ($t < 200$) | 33.32 | 22.1578 | 2.6680 |
| Random (20% steps) | 33.10 | 22.1247 | 2.7180 |
| Full steps (Default) | 33.52 | **22.2704** | **2.7187** |

*Table 9.* **Ablation on perturbation scale dynamics.** The performance of applying GAD with timestep-adaptive perturbation scale $h$ on PixArt-$\alpha$.

| Scale Schedule $h(t)$ | CLIP ↑ | PickScore ↑ | Vendi ↑ |
|---|---|---|---|
| Linear (decreasing) | 33.32 | 22.2166 | 2.6694 |
| Linear (increasing) | **33.53** | 22.1427 | **2.7202** |
| Fixed (Ours: $h = 10^{-2}$) | 33.52 | **22.2704** | 2.7187 |

sitivity of geometric alignment, we ablate the timestep intervals where the GAD objective is applied. As shown in Tab. 8, applying GAD exclusively to the middle noise levels ($t \in [200, 600]$) restores a significant portion of generative diversity (Vendi score 2.7140 vs. baseline 2.5967). Interestingly, high-noise supervision ($t > 600$) yields the best semantic alignment, while middle-noise supervision is most critical for structural diversity. This temporal disentanglement reveals that GAD does not require dense trajectory alignment to be effective. Notably, the sparse supervision strategy (e.g., mid-noise only) achieves performance highly competitive with the full-step setting, while incidentally offering a significant reduction in the computational overhead associated with the finite-difference approximation.

**Perturbation Scale Dynamics.** We ablate the fixed scale against timestep-adaptive schedules in Tab. 9. The results indicate that an increasing linear schedule (i.e., assigning a smaller $h$ at high noise and a larger $h$ at low noise) further enhances both the CLIP score and Vendi diversity. This aligns with the intuition of the reverse generation process: at early stages (high noise), minute perturbations are exponentially amplified by subsequent sampling steps, requiring a finer scale to maintain local linearity. At later stages (low noise), the model focuses on detail refinement and exhibits greater robustness, allowing a larger $h$ to capture local geometric variations more effectively.

## 6. Conclusion

We identified and formalized *noise sensitivity degradation* in diffusion distillation, showing that standard pointwise objectives preserve perceptual quality but fail to retain the teacher's local functional geometry, leading to reduced controllability and mode collapse. To address this issue, we proposed Geometry-Aware Distillation (GAD), which aligns teacher and student models through Jacobian-vector product matching, explicitly preserving their response to input perturbations. Extensive experiments across multiple architectures and distillation paradigms demonstrate that GAD consistently improves functional alignment. Importantly, restoring noise sensitivity translates into practical gains, including improved layout control, enhanced generative diversity, and more effective transfer of test-time optimization, without compromising image fidelity.

**Limitations and Future Work.** While GAD incurs zero inference cost, our finite-difference approximation introduces additional training overhead (detailed in Appendix B), as it requires an extra forward pass. However, this remains more efficient than standard second-order methods by reusing cached conditions. Future work could explore more sample-efficient estimators for Jacobian alignment or investigate the theoretical connections between geometric sensitivity and generalization bounds in generative modeling. Furthermore, extending GAD to video diffusion distillation, where temporal consistency relies heavily on noise correlations, presents a promising direction.

## Acknowledgments

This work was partially supported by the National Natural Science Foundation of China under Grant 62372341. YZ was also partially supported by the Hi! PARIS initiative and the ANR/France 2030 program (ANR-23-IACL-0005). Additional support was provided by the MiLM Plus Team at Xiaomi Inc.

## Impact Statement

This work aims to improve accelerated diffusion models by addressing a critical limitation in existing distillation approaches: the loss of sensitivity to initial noise. By preserving this property during acceleration, our method allows fast generative models to retain the diversity, controllability, and stochastic expressiveness of their teacher counterparts, rather than converging to high-quality yet overly homogeneous outputs. As with many advances in AI-based generative modeling, this work may have broader societal implications that are common to the deployment of such technologies. From a technical perspective, however, we do not foresee any distinct or additional risks arising from our method beyond those generally associated with current text-to-image generative models.

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

# Appendices

The appendix is structured as follows: First, Sec. A provides comprehensive ablation studies and parameter settings, analyzing the method's sensitivity to key hyperparameters like the perturbation scale and weighting parameter. Next, Sec. B evaluates the computational complexity and training efficiency of our proposed GAD method across different backbones. Sec. C details the representative baseline distillation methods used for comparison, while Sec. D provides detailed implementation specifications to ensure reproducibility. Furthermore, Sec. E presents additional quantitative evaluations focused on zero-shot FID scores. Finally, Sec. F offers a 2D toy example using the Swiss Roll dataset to intuitively visualize the geometric gap, and Sec. G includes more qualitative visualization results demonstrating improvements in diversity and noise-based layout control.

## A. Ablation Study and Parameter Settings

**Experimental Setup.** We evaluate the sensitivity of GAD to key hyperparameters using PixArt-$\alpha$ (DiT). All ablation experiments are conducted over 2k training steps. To monitor performance dynamically, metrics are computed online using 50 randomly sampled MS-COCO prompts, with 4 images generated per prompt to calculate intra-prompt LPIPS.

**Sensitivity to Perturbation Scale $h$.** We investigate the impact of the finite difference interval $h \in \{10^{-5}, \ldots, 1\}$. As illustrated in Fig. 7, the choice of $h$ involves a critical trade-off between capturing structural trends and maintaining local linearity. When $h$ is too small (e.g., $10^{-5}$), the regularization signal becomes numerically insignificant, resulting in a collapse toward repetitive modes as indicated by the lowest intra-prompt LPIPS (green line). Conversely, an excessively large scale (e.g., $h = 1$) violates the local tangent space assumption, degrading the learned manifold and leading to sub-optimal fidelity. We select $h = 10^{-2}$ (grey line) as the default, as it provides the most stable convergence and an optimal equilibrium between curvature preservation and generation quality.

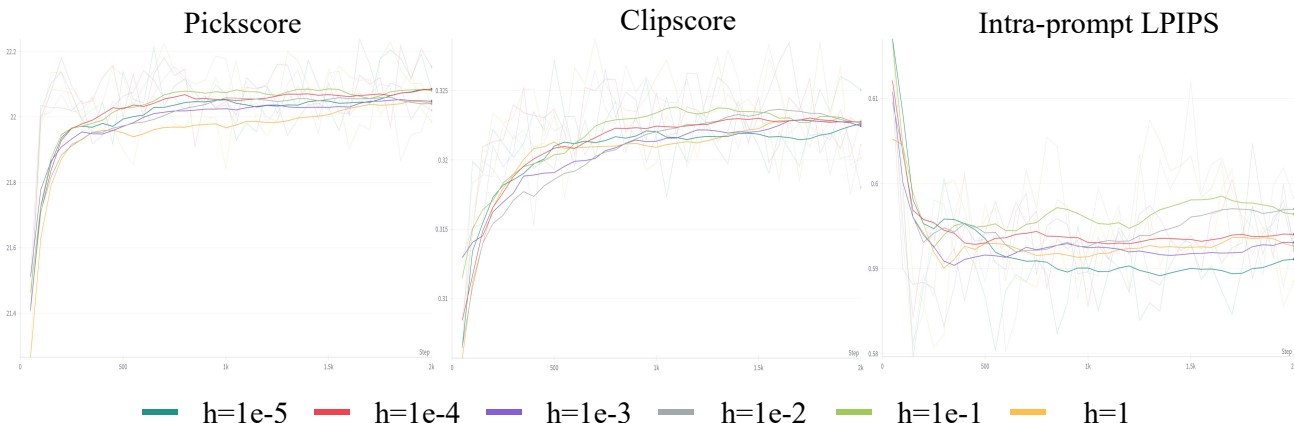

*Figure 7.* **Ablation study on the perturbation scale** $h$. Training curves on PixArt-$\alpha$ for Pickscore (left), CLIP Score (middle), and Intra-prompt LPIPS (right). The results indicate that very small $h$ values fail to restore diversity (LPIPS), while $h = 10^{-2}$ (grey) achieves an optimal equilibrium between structural sensitivity and generation quality.

**Sensitivity to Weighted Parameter $\lambda$.** We further analyze the balance between the base distillation loss and geometric regularization by varying $\lambda \in \{0.1, \ldots, 3.0\}$. GAD demonstrates strong robustness across this range (Fig. 8). Specifically, an excessively large weight (e.g., $\lambda = 3.0$) slightly interferes with the primary pointwise objective, causing a marginal decline in PickScore and CLIP Score. In contrast, a small weight (e.g., $\lambda = 0.1$) provides insufficient regularization to fully restore the model's noise sensitivity, leading to diminished LPIPS. The setting $\lambda = 1.0$ (grey line) serves as the optimal balance, effectively reconciling the teacher's local geometry with high semantic alignment.

## B. Complexity and Training Efficiency

To evaluate the computational footprint of GAD, we analyze the training overhead across three representative distillation settings. As GAD employs a finite-difference approximation, it primarily introduces one additional forward pass per training step. Importantly, the gradient is only back-propagated through the student model, and the teacher/auxiliary estimators remain in constant memory during the GAD loss computation.

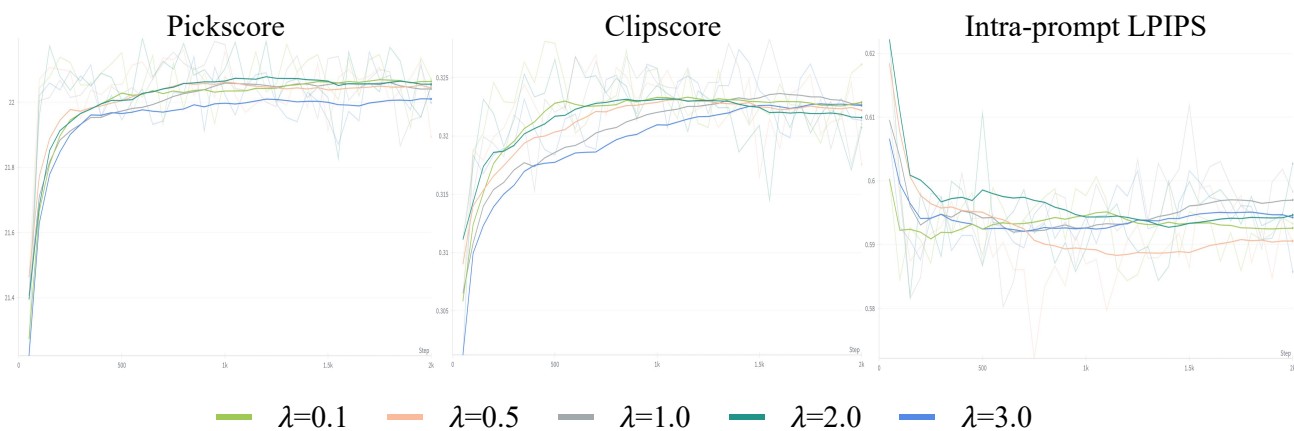

*Figure 8.* **Ablation study on the weighting parameter** $\lambda$. Training curves on PixArt-$\alpha$ for Pickscore (left), CLIP Score (middle), and Intra-prompt LPIPS (right). Metrics are computed online during training on a subset of 50 COCO prompts. The results highlight a trade-off: high $\lambda$ values slightly compromise fidelity scores, while low $\lambda$ values lead to diminished LPIPS (diversity), with $\lambda = 1.0$ yielding the most balanced performance.

In Table 10, we report the wall-clock time per iteration and peak GPU memory usage during training for the baseline methods versus their GAD-enhanced versions. Benchmarks for LADD (SD2.1) were performed on NVIDIA A800 (80GB) GPUs, while TDM (PixArt-$\alpha$) and SiD (SANA) were benchmarked on NVIDIA RTX 4090 (48GB) GPUs.

*Table 10.* **Detailed computational overhead.** We compare the baseline training cost with the GAD-integrated version. Time refers to the wall-clock duration per training iteration. Memory refers to the peak VRAM usage per GPU during training.

| Method (Model) | Resolution | Training Time (s / iter) | | Peak Memory (GB) | |
|---|---|---|---|---|---|
| | | Baseline | + GAD (%) | Baseline | + GAD (%) |
| LADD (SD2.1) | $512^2$ | 20.89 | 33.77 (+61.66%) | 47.45 | 69.67 (+46.82%) |
| TDM (PixArt-$\alpha$) | $512^2$ | 4.38 | 5.97 (+36.30%) | 31.44 | 32.61 (+3.72%) |
| SiD (SANA) | $512^2$ | 48.34 | 80.49 (+66.51%) | 34.08 | 40.31 (+18.28%) |

The empirical results demonstrate that GAD introduces a marginal overhead. The relative increase in training time is significantly lower than a factor of $2\times$ (which a naive second-order method would require), because the perturbation $\Delta\epsilon$ reuses the cached conditions and latents from the primary forward pass. Moreover, since GAD is only active during training, it incurs **zero additional cost** during inference.

## C. Details of Baseline Distillation Methods

To comprehensively evaluate the proposed GAD, we compare against a diverse set of representative diffusion distillation paradigms spanning three major backbones: Stable Diffusion v2.1 (U-Net), PixArt-$\alpha$ (DiT), and SANA (Flow-based DiT). These baselines cover adversarial distillation, consistency modeling, distribution matching, and trajectory-level alignment, reflecting the dominant directions in recent diffusion acceleration research.

### C.1. Distillation for SD2.1 (U-Net)

- **SD-Turbo (Sauer et al., 2024b):** SD-Turbo is trained using *Adversarial Diffusion Distillation (ADD)*, which combines two primary objectives: (i) an adversarial loss that utilizes a discriminator to ensure the generated samples align with the manifold of real images, and (ii) a distillation loss that leverages a frozen teacher diffusion model to maintain the original model's extensive knowledge. By integrating these losses, SD-Turbo achieves high-quality one-step generation without the need for classifier-free guidance during inference. In our experiments, we use the publicly available checkpoint from Hugging Face[1].

---

[1] https://huggingface.co/stabilityai/sd-turbo

- **SwiftBrush & SwiftBrushv2 ([Nguyen & Tran, 2024](); [Dao et al., 2024]()):** SwiftBrush formulates distillation as *variational score distillation*, transferring a pretrained text-to-image prior into a lightweight student by matching predicted noise distributions under a variational objective. SwiftBrushv2 further refines the objective with improved initialization and stability techniques, emphasizing faithful reconstruction of teacher outputs in few-step regimes. For our evaluation, we employ the publicly released checkpoints of SwiftBrush[2] and SwiftBrushv2[3].

- **Trajectory Consistency Distillation (TCD) ([Zheng et al., 2024]()):** TCD extends consistency models by explicitly enforcing *multi-step trajectory consistency* between intermediate denoising states. The student is trained such that predictions at different timesteps remain coherent under teacher-guided transitions, improving sample quality for small numbers of sampling steps while implicitly regularizing the temporal structure of the diffusion process. In our experiments, we use the publicly available TCD-SD21-base LoRA checkpoint[4].

- **LADD ([Sauer et al., 2024a]()):** Latent Adversarial Diffusion Distillation introduces adversarial supervision in the *latent space* rather than pixel space in adversarial diffusion distillation, using a strong teacher to provide high-level semantic feedback. By aligning latent distributions through a discriminator, LADD prioritizes semantic fidelity and perceptual realism. In our study, LADD serves as the foundational framework for implementing GAD on SD2. To ensure strict variable control and a fair comparison, we retrained the models both with and without GAD regularization, utilizing the publicly available Nitro-1 codebase[5].

## C.2. Distillation for PixArt-$\alpha$ (DiT)

- **Latent Consistency Models (LCM) ([Luo et al., 2023]()):** LCM trains the student to map any intermediate latent state along the probability flow ODE directly to the clean data point. This formulation collapses the entire denoising trajectory into a single consistency constraint, allowing flexible few-step sampling while sacrificing explicit modeling of trajectory geometry. In our evaluation, we utilize the publicly available PixArt-LCM-XL-2 checkpoint[6].

- **Distribution Matching Distillation (DMD) ([Yin et al., 2024b]()):** DMD frames distillation as minimizing the KL divergence between student-generated samples and the real data distribution. It employs two score estimators-one from a pretrained diffusion model and one from a learned critic-to approximate the density ratio, enabling few-step generation without requiring paired teacher trajectories. In our experiments, we use the publicly available PixArt-Alpha DMD checkpoint[7].

- **FLASH ([Chadebec et al., 2025]()):** Flash Diffusion (FLASH) is a versatile distillation method designed to maintain high image quality under drastic step reduction. The student is trained to predict, in a single step, a multi-step denoised output from a teacher model. This process is supervised by a combination of: (i) a distillation loss between student and teacher predictions, (ii) an adversarial objective to drive the student distribution toward the real image manifold, and (iii) a distribution matching (DMD) loss to prevent drifting from the teacher's learned distribution. For our experiments, we utilize the publicly released FLASH-PixArt checkpoint[8].

- **YOSO ([Luo et al., 2025a]()):** YOSO (*You Only Sample Once*) is a one-step generation framework that integrates diffusion processes with GANs while addressing the training instability and mode collapse common in adversarial distillation. The core mechanism is a self-cooperative learning strategy that smoothes the adversarial divergence by using the denoising generator itself: it treats one-step generation from less corrupted samples as the ground truth to supervise the generation from more corrupted samples. For text-to-image synthesis, YOSO incorporates several specialized techniques, including latent perceptual loss, Informative Prior Initialization (IPI), and a quick adaptation stage to refine the noise scheduler. In our experiments, we utilize the publicly available YOSO-PixArt-512 checkpoint[9].

- **Trajectory Distribution Matching (TDM) ([Luo et al., 2025b]()):** TDM aligns the *entire sampling trajectory distribution* of the student with that of the teacher by matching probability flow ODE paths. By enforcing distribution consistency

---

[2] https://huggingface.co/thuanz123/swiftbrush
[3] https://drive.google.com/drive/folders/1eUVwTrkOVWT2gCJ4TiWlZmCV2sODuvQD
[4] https://huggingface.co/h1t/TCD-SD21-base-LoRA
[5] https://github.com/AMD-AGI/Nitro-1
[6] https://huggingface.co/PixArt-alpha/PixArt-LCM-XL-2-1024-MS
[7] https://huggingface.co/PixArt-alpha/PixArt-Alpha-DMD-XL-2-512x512
[8] https://huggingface.co/jasperai/flash-pixart
[9] https://huggingface.co/Luo-Yihong/yoso_pixart512

across timesteps, TDM aims to preserve global diffusion behavior beyond marginal sample quality. In our study, TDM serves as the foundational distillation framework for our experiments on PixArt-$\alpha$. To ensure a fair comparison and strict variable control, we utilized the official TDM codebase[10] to retrain the models both with and without GAD regularization.

### C.3. Distillation for SANA (Flow-based DiT)

- **Score Identity Distillation (SiD) (Zhou et al., 2025b):** SiD provides a unified distillation framework for both Gaussian diffusion and flow matching by leveraging a simplified derivation based on Bayes' rule and conditional expectations. It demonstrates that score distillation can be applied broadly to text-to-image flow-matching models without requiring teacher fine-tuning or architectural changes. In our study, SiD serves as the foundational distillation framework for implementing GAD on SANA. To ensure strict variable control and a fair comparison, we utilized the official SiD codebase[11] to retrain the distillation models both with and without GAD regularization.

## D. Detailed Implementation Specifications

This section provides additional technical details regarding the network architectures, hyperparameter settings, and the training procedure to ensure reproducibility. Across all distillation settings, we employ the AdamW optimizer with $\beta_1 = 0.9$ and $\beta_2 = 0.999$. The balancing weight $\lambda$ is set to 1.0 across all models. Specific parameters for each backbone are summarized in Table 11.

*Table 11.* **Key training hyperparameters.** Detailed configurations for GAD integration across different backbones.

| Configuration | SD2.1 (U-Net) | PixArt-$\alpha$ (DiT) | SANA (Flow-based DiT) |
|---|---|---|---|
| Learning Rate | $1 \times 10^{-6}$ | $4 \times 10^{-6}$ | $5 \times 10^{-6}$ |
| Total Iterations | 100K | 2K | 5K |
| Batch Size (per GPU) | 64 | 64 | 8 |
| Precision | BF16 | FP16 | BF16 |
| Perturbation Scale $h$ | 0.0001 | 0.01 | 0.01 |
| GAD Weight $\lambda$ | 1.0 | 1.0 | 1.0 |

## E. Additional Quantitative Evaluation: FID Scores

In Sect. 5.2 of the main text, we evaluate the general generation quality using PickScore and CLIP Score to assess image fidelity and text-image alignment. To further address how the models fit the overall data distribution, we report the zero-shot Fréchet Inception Distance (FID) on the MS-COCO 30K dataset in Tab. 12.

The results demonstrate that GAD consistently enhances distributional alignment with the teacher model, evidenced by the lower "FID vs. Teacher" scores across all architectures. Notably, in configurations where the baseline distillation method (e.g., TDM on PixArt-$\alpha$) happens to achieve a lower FID than the teacher itself, adding GAD leads to a marginal increase in the "FID vs. COCO-30k" metric. This phenomenon aligns with our core objective: GAD prioritizes capturing the teacher's underlying geometric distribution rather than merely optimizing for external dataset metrics. Consequently, the student's output distribution shifts to be more "teacher-like," pulling the FID closer to the teacher's original performance. This further validates that GAD effectively restores the original generative dynamics and distribution of the teacher model.

## F. 2D Toy Example: Swiss Roll Visualization

To intuitively visualize the "geometric gap" and the structural degradation caused by standard pointwise distillation, we design a 2D toy experiment using the classic Swiss Roll dataset. The Swiss Roll represents a highly curved, low-dimensional

---

[10] https://github.com/Luo-Yihong/TDM
[11] https://github.com/apple/ml-sid-dit/

*Table 12.* **Quantitative comparison of zero-shot FID scores on the MS-COCO 30K dataset.** Adding GAD consistently reduces the FID discrepancy between the distilled student and the teacher model.

| Model | Method | FID vs. COCO-30k ↓ | FID vs. Teacher ↓ |
|---|---|---|---|
| **SD2** | Teacher | 16.5505 | - |
| | LADD | 16.7400 | 11.2920 |
| | LADD + Ours | **16.5567** | **7.8463** |
| **PixArt-$\alpha$** | Teacher | 28.4923 | - |
| | TDM | **25.3909** | 5.8014 |
| | TDM + Ours | 26.3235 | **5.1359** |
| **SANA** | Teacher | 26.5129 | - |
| | SiD | 28.6817 | 8.4830 |
| | SiD + Ours | **27.2026** | **7.3596** |

data manifold embedded in a 2D space, making it an ideal testbed for evaluating whether a model preserves complex geometric structures.

We first train a standard diffusion model on the Swiss Roll dataset to serve as the Teacher model, which samples using 40 DDIM steps. Then, we distill this teacher into a 4-step Student model using a standard pointwise distillation objective, and compare it against a 4-step Student trained with our Geometry-Aware Distillation (GAD).

As illustrated in Fig. 9, the Teacher model successfully learns the highly curved data distribution. However, the standard Student model struggles to capture the continuous curvature. Because its pointwise objective optimizes for averaged responses, it attempts to map the distribution via erroneous "shortcuts" (highlighted by the red boxes). These shortcuts result in generated points falling into low-density regions that are entirely outside the original data manifold, causing significant distribution shifts.

In contrast, the Student trained with GAD eliminates these structural shortcuts. By explicitly aligning the Jacobian-vector products (JVP), GAD forces the student to respect the local curvature and directional gradients of the teacher's mapping. Consequently, the GAD Student preserves the intricate geometry of the original manifold, yielding a distribution that is consistent with both the Teacher model and the ground-truth training data.

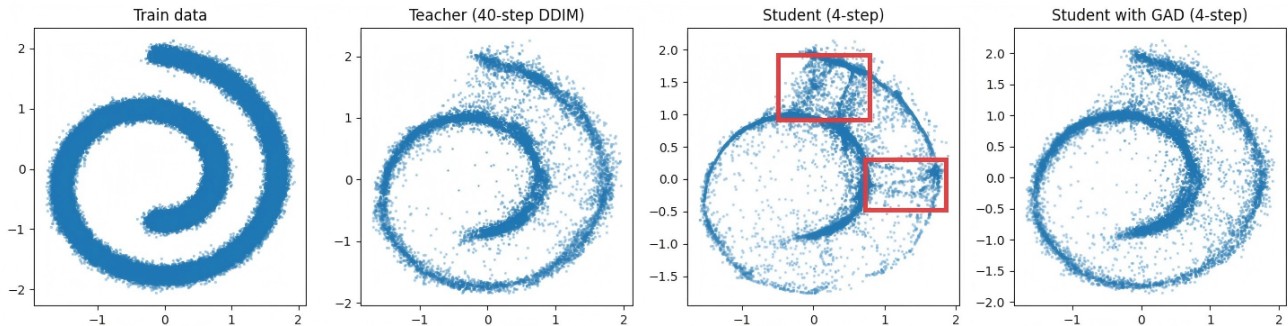

*Figure 9.* **A Swiss Roll toy example visualizing the restoration of geometry.** Left to right: Ground truth training data, Teacher model (40-step DDIM), standard Student (4-step), and our GAD Student (4-step). Standard distillation leads to structural "shortcuts" (red boxes) across the complex curves, causing severe distribution shifts. In contrast, GAD accurately preserves the teacher's original geometry and manifold curvature.

# G. More Visualization Results

*An old victorian style bed frame in a bedroom.*

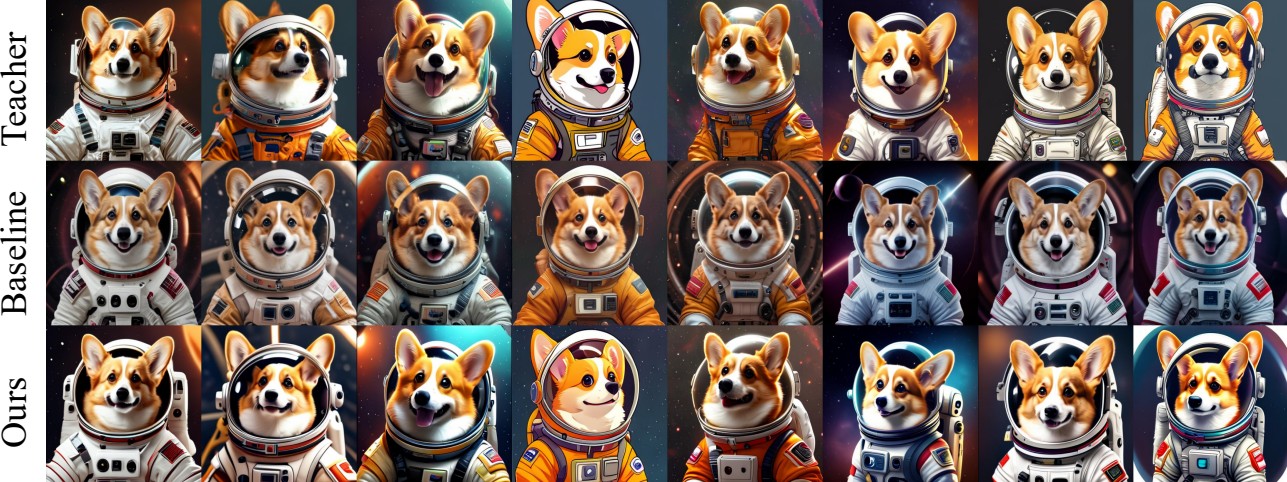

*A cute corgi in a spacesuit.*

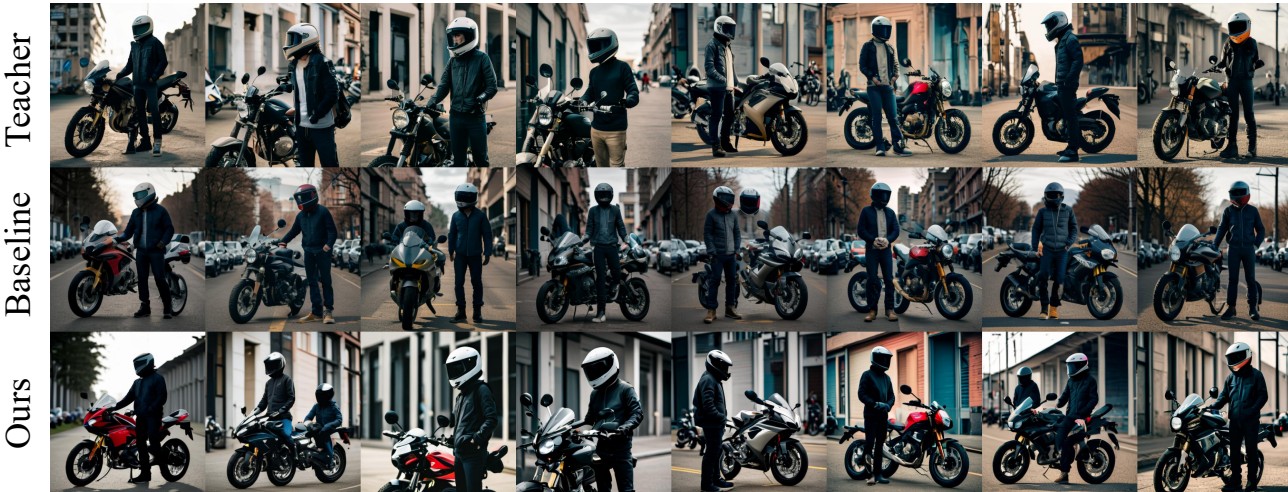

*A person in a helmet standing by their motorcycle.*

*Figure 10.* **More visualization of diversity improvement.** The experiments are conducted on the SANA model using the Score Identity Distillation (SiD) as the foundational distillation framework.

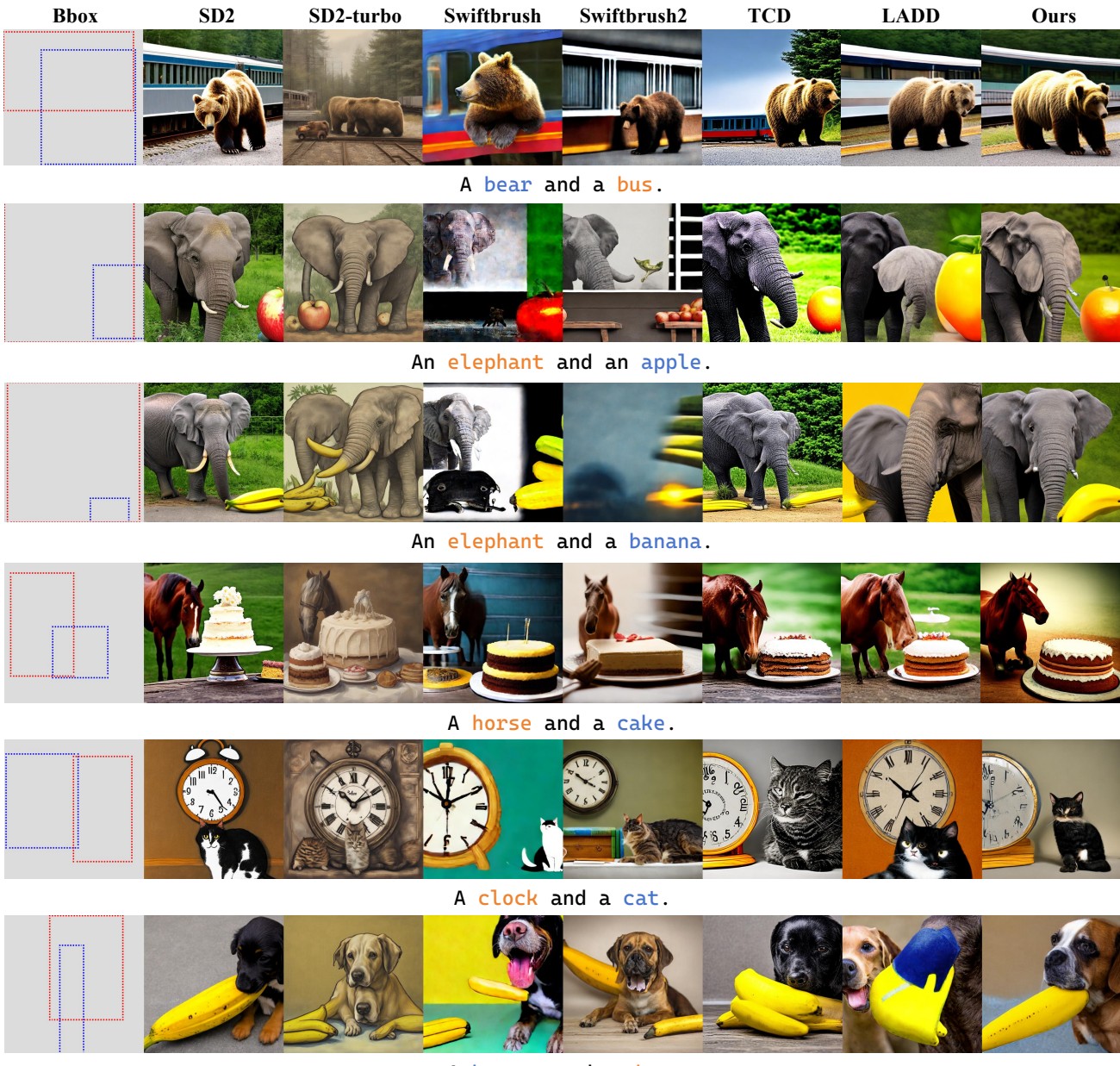

*Figure 11.* **More visualization of noise-based layout control.** The experiments are conducted on the Stable Diffusion v2 (SD2) model using LADD as the foundational distillation framework.

