# OpenReview forum: "Restoring Initial Noise Sensitivity in Text-to-Image Distillation through Geometric Alignment"
_ICML.cc/2026/Conference — ICML 2026 regular_

### Official Review · Reviewer_5G7g · 2026-03-06

**Soundness:** 3
**Presentation:** 3
**Significance:** 3
**Originality:** 2
**Overall Recommendation:** 3
**Confidence:** 2

**Summary:**

This paper studies an observed loss of initial noise sensitivity in distilled text-to-image models, arguing that common distillation objectives reduce the student model’s responsiveness to seed perturbations, which negatively affects diversity and noise-driven controllability. The authors frame this as a loss of local geometric alignment between teacher and student with respect to the input noise.
To address this, they propose Geometry-Aware Distillation (GAD), which aligns teacher and student local responses to small noise perturbations using a finite-difference approximation of Jacobian–vector products. The method is designed to be plug-and-play with existing distillation frameworks and is evaluated across multiple models and tasks, showing improved diversity and controllability at the cost of increased training overhead.

**Compliance With Llm Reviewing Policy:**

Affirmed.

**Key Questions For Authors:**

See weaknesses.

**Limitations:**

The authors acknowledge practical limitations such as increased training overhead and sensitivity to hyperparameters. However, the discussion does not sufficiently address key methodological limitations. In particular, the paper does not explicitly discuss the reliance on proxy metrics (e.g., seed identifiability and downstream tasks) instead of directly measuring teacher-student geometric alignment. The absence of direct Jacobian-based validation and limited causal analysis of the source of noise sensitivity degradation are important constraints that should be more clearly acknowledged.
A more explicit and structured discussion of these conceptual limitations would improve transparency and help readers better understand the scope and boundaries of the contribution.

**Strengths And Weaknesses:**

## Strength:
The paper addresses a practically relevant and underexplored issue in text-to-image distillation: the loss of sensitivity to initial noise after reducing inference steps. Framing initial noise sensitivity as an important property for diversity and controllability is meaningful, particularly in interactive generation settings where users rely on seed variation and latent exploration. This problem formulation reflects real usage scenarios and gives the work practical significance.
The proposed method is technically reasonable and straightforward. Extending standard distillation objectives with a finite-difference approximation of Jacobian–vector products is mathematically sound and does not rely on unrealistic assumptions. The implementation is compatible with black-box teachers and does not require architectural changes, making it easy to integrate into existing pipelines. This simplicity and modularity enhance its practical value and reproducibility.
Empirically, the evaluation covers multiple distillation frameworks and base models, and examines diversity, controllability, alignment metrics, and computational overhead. The inclusion of ablations on perturbation scale and regularization strength adds credibility to the analysis. Training cost increases are transparently reported. The overall experimental setup appears methodologically appropriate and sufficiently detailed for reproduction.
From an originality perspective, while derivative matching is not a new concept, applying local geometry alignment specifically to preserve seed-dependent behavior in distilled text-to-image models represents a creative and domain-specific combination of existing ideas. The paper highlights a tradeoff between efficiency and latent controllability that may encourage further investigation in generative model distillation.
The paper is generally well structured and readable. The motivation is clearly stated, the method is explained concisely, and diagrams help illustrate the intuition behind pointwise versus geometry-aware alignment.

## Weaknesses:
1. The core claim of restoring teacher–student geometric alignment is not directly validated. The paper relies on proxy metrics rather than reporting direct measurements of JVP similarity, Jacobian norms, or spectral alignment.
2. The causal source of the degradation in noise sensitivity is not clearly disentangled. It remains unclear whether the issue stems from reduced NFE, the specific distillation objective, architectural bottlenecks, or their interaction.
3. The methodological novelty is moderate. The approach closely resembles prior Jacobian or Sobolev-style derivative matching methods, and the distinction from existing literature should be clarified more explicitly.
4. Some evaluation metrics, such as seed identifiability and NoiseQuery transfer, are not standard or fully justified proxies for geometric restoration. Their relationship to true initial noise sensitivity requires clearer theoretical or empirical grounding.
5. The connection between the claimed restoration of local geometry and the selected downstream tasks is suggestive but not rigorously established. Stronger intermediate evidence linking geometry to performance would improve the argument.
6. The training overhead introduced by additional forward passes is non-trivial. A more thorough cost–benefit analysis would clarify whether the gains justify the added complexity.
7. The intuitive explanation that pointwise distillation “flattens the landscape” is not theoretically analyzed. A clearer formal argument or controlled toy example would strengthen the soundness of this claim.
8. Figure 3 provides useful intuition but does not accurately or completely reflect the formal definition of $L_{\text{GAD}}$ . Clarifying how the visualization maps to the finite-difference objective would improve precision and reduce potential misunderstanding.

---

> ### Author Rebuttal · Authors · 2026-03-31
>
> > W1/W5/L1: Direct Geometric Alignment Metrics.
>
> Following the suggestion, we further report direct geometric metrics below. The results consistently demonstrate that GAD improves geometric alignment between student and teacher, narrowing the gap in local responses.
>
> | Method | JVP Cos Similarity $\uparrow$ | Jacobian Norm Ratio $\uparrow$ | Spectral KL $\downarrow$ | JVP MSE $\downarrow$ |
> |-|-| -| - | - |
> |Teacher Model |1.000|1.000|0.000|0.000|
> |TDM | 0.012 |0.98|0.008|0.0003|
> |TDM + GAD (Ours) |**0.014**| **0.99** | **0.006** | **0.0002** |
>
>
> > W2/L2: Causal Source of Sensitivity Degradation.
>
> We view causal analysis as an upstream problem, whereas our work focuses on a complementary downstream solution: restoring sensitivity after degradation is observed. Investigating all the causal origins, while valuable, is largely orthogonal to our scope and could inspire alternative solutions in future work. Nevertheless, we provide the following insights:
> - **Reduced NFE**: Reducing teacher NFE alone does not show a monotonic drop in diversity (Vendi: 2.7343 (20 steps), 4.0264 (4 steps), and 1.8497 (2 steps)). In contrast, increasing the distillation intensity consistently degrades sensitivity (see Q1, BEpg), indicating the issue is mainly tied to the distillation process.
> - **Architecture**: Our experiments across UNet, DiT, and Flow-DiT prove this collapse is a universal phenomenon.
> - **A general solution (distillation objective).** GAD offers a general, model-agnostic solution by directly aligning Jacobian-vector products without changing architecture, capacity, or the base distillation objective.
>
> > W3: Distinction from Jacobian/Sobolev Methods.
>
> - **New Problem:** Unlike prior Jacobian/Sobolev works aimed at general knowledge transfer and efficient training, we are the first to identify and formalize the "sensitivity collapse" specifically in diffusion acceleration.
> - **Scalability:** Standard Sobolev methods are prohibitive for high-resolution images ($d \approx 10^5$). GAD’s finite-difference scheme reuses cached conditions, enabling second-order consistency on industrial models.
> - **Functional Control:** By restoring noise sensitivity, GAD empowers distilled models with robust downstream controllability and enhanced diversity—capabilities that remain limited for standard distillation paradigms.
>
>
> > W4: Validity of Proxy Metrics.
>
> - From a first-order perspective, output variation under noise perturbation is governed by the Jacobian: $\Phi(z + \epsilon v) - \Phi(z) \approx J_\Phi(z) v$. Seed identifiability measures whether different noise vectors $v$ map to distinct regions in the output, which depends on the full-rankness and orientation of $J_\Phi$​. Similarly, NoiseQuery requires preserving  teacher’s directional gradients (Jacobian) for consistent noise-to-image mapping across models.
> - As shown in our W1 response, GAD improves both proxies and direct geometric metrics.
> - To further substantiate the validity of our proxies, we calculate the Pearson Correlation Coefficient ($r$) between JVP MSE and Seed Identifiability across 15 different training checkpoints. A strong **negative correlation of $\mathbf{r = -0.89}$** confirms our proxies are faithful indicators of underlying Jacobian alignment.
>
> > W6: Cost–benefit analysis.
>
> Please refer to Reviewer hCwK L1.
>
> > W7: Formal Analysis of Landscape Flattening.
>
> The flattening effect of pointwise distillation is well-documented in the literature.
> - Knowledge Distillation. Pointwise KD acts as instance-specific label smoothing [1], which reduces the Lipschitz constant (an upper bound on input-output variation) [2]. Self-distillation terms (with the same architectures) also flatten the loss landscape and dampen gradient responses [3].
> - Diffusion Distillation. Pointwise matching often leads to over-smoothing [4,5], where the student approximates a conditional expectation (mean) rather than a precise manifold. GAD’s JVP alignment explicitly restores the teacher's differential response and curvature.
>
> > W8: Figure 3 vs. Finite-Difference Objective.
>
> The visualization in Fig.3 explains how $L_{\text{GAD}}$ restores noise sensitivity by counteracting density collapse. Standard distillation enforces $\Phi_S(z) \to \Phi_T(z)$, but often causes outputs to collapse toward high-density regions (darker areas in Fig 3). This makes student responses significantly smaller than the teacher's ($||\Delta \Phi_S|| \ll ||\Delta \Phi_T||$). GAD provides a counter-force by aligning the differences ($\Delta \Phi_S \approx \Delta \Phi_T$), ensuring the student preserves the local stretching and orientation of the teacher’s manifold.
>
> [1] Self-Distillation as Instance-Specific Label Smoothing. NeurIPS 2020.
> [2] Lipschitz Continuity Guided Knowledge Distillation. ICCV 2021.
> [3] Revisiting Self-Distillation. Arxiv 2206.
> [4] Score Distillation via Reparametrized DDIM. NeurIPS 2024.
> [5] One-step Diffusion with Distribution Matching Distillation. CVPR 2024.

---

> > ### Author Rebuttal · Reviewer_5G7g · 2026-04-01
> >
> > I appreciate the authors’ effort in strengthening the paper through the rebuttal, particularly by including direct geometric metrics. This makes the empirical claims clearer and improves the overall rigor of the work.
> >
> > However, I still have a few concerns that I would like the authors to clarify.
> >
> > First, I find the causal explanation of sensitivity degradation somewhat unclear. The paper attributes this issue mainly to pointwise distillation objectives, but it remains ambiguous how much of the effect is due to the objective itself versus other factors such as reduced NFE, architectural constraints, or their interaction. It would be helpful if the authors could provide a more controlled analysis (e.g., fixing NFE while varying objectives, or vice versa) to better isolate the primary source.
> >
> > Second, I am not fully convinced about the interpretation of seed identifiability as a proxy for meaningful sensitivity. While it is intuitive that different seeds should lead to different outputs, it is unclear whether higher identifiability necessarily corresponds to semantically meaningful or controllable variation, rather than capturing superficial or model-specific artifacts. Additional clarification or supporting analysis would strengthen this point.
> >
> > Finally, Figure 3 provides helpful intuition, but I believe that the connection to the actual $L_\text{GAD}$ objective could be made clearer. It would be beneficial to more explicitly show how the visualization relates to the finite-difference formulation used in training.
> >
> > If these points are clarified more convincingly, I would be open to revisiting my assessment.

---

> > > ### Author Response · Authors · 2026-04-06
> > >
> > > > Q1: Causal explanation of sensitivity degradation.
> > >
> > > To isolate the effect of the distillation objective from the number of sampling steps (NFE), we conduct a controlled study on PixArt by fixing the teacher and varying both the objective (standard pointwise vs. GAD) and NFE. Vendi scores are shown below:
> > >
> > > |Method|1-step$\uparrow$|2-step$\uparrow$|4-step$\uparrow$|
> > > |-|-|-|-|
> > > |Baseline (TDM)|2.05|2.61|2.59|
> > > |GAD (Ours)|**2.09**|**2.67**|**2.72**|
> > >
> > > - Changing NFE also affects sensitivity. In particular, extreme compression to 1-step generation leads to lower sensitivity, as model capacity is strained when compressing complex multi-step trajectories into single- or few-step mappings.
> > > - At any fixed NFE, GAD consistently outperforms the baseline.
> > > - GAD benefits more from additional steps, suggesting a positive interaction between the geometric regularizer and longer trajectories.
> > >
> > > Overall, both the distillation objective and NFE affect model sensitivity. However, increasing NFE incurs higher inference cost, whereas GAD, by modifying the distillation objective, enables higher sensitivity without increasing inference cost.
> > >
> > > > Q2: Interpretation of seed identifiability.
> > >
> > > - We thank the reviewer for this thoughtful comment and agree that seed identifiability **alone** does not **guarantee** meaningful semantic variation.
> > > - Regarding the relationship between seed identifiability and meaningful sensitivity: **we consider it a supporting factor for meaningful variation (jointly achieving high semantic diversity and strong text alignment), but does not alone determine it**. To examine this, we report seed identifiability together with semantic diversity (Vendi) and text alignment (CLIP) across different methods in the table above. We observe a consistent trend that **models with low seed identifiability tend to struggle to jointly achieve high semantic diversity and strong text alignment, whereas higher identifiability generally facilitates achieving both metrics at high levels simultaneously**. SwiftBrush exhibits significantly lower identifiability and weaker diversity and alignment, suggesting insufficient responsiveness to input noise limits semantic diversity. In contrast, methods with higher identifiability achieve better overall performance. Meanwhile, TCD attains high diversity despite slightly lower identifiability than LADD, but with reduced CLIP scores, suggesting that part of the diversity may arise from weaker semantic alignment rather than meaningful variation.
> > >
> > > |Method|Seed identifiability$\uparrow$|Vendi$\uparrow$|CLIP$\uparrow$|
> > > |-|-|-|-|
> > > |Teacher|93.70%|3.93|29.72|
> > > |SwiftBrush|52.90%|3.85|26.41|
> > > |TCD|87.30%|4.11|26.41|
> > > |LADD|87.60%|3.90|27.21|
> > > |GAD (Ours)|92.40%|3.98|27.52|
> > >
> > > - Our method (GAD) does **not optimize** this metric during training; instead, it improves sensitivity by aligning the local geometry of the teacher’s generative manifold, inheriting the teacher’s meaningful noise-to-image mapping. Seed identifiability is only used as a **zero-shot evaluation metric**. Overall, while insufficient alone, it serves as a useful supporting indicator when interpreted alongside semantic diversity and alignment.
> > >
> > >
> > > > Q3: Connection between Fig.3 and the objective.
> > >
> > > Figure 3 strictly corresponds to the finite-difference form in Eq. 6. We add an explicit mapping:
> > >
> > > |Fig.3 Element|Math|Description|
> > > |-|-|-|
> > > |Input `z` (left: gray rectangle)|$\mathbf{z}$|Original input|
> > > |Perturbed input `z'` (left: gray rectangle)|$\mathbf{z}' = \mathbf{z} + h\mathbf{v}$|Finite-difference perturbed input|
> > > |Perturbation arrow z → z' (left: gray circle)|$h\mathbf{v}$|Perturbation along random direction $\mathbf{v} \sim \mathcal{N}(0, I)$|
> > > |Teacher output at z, z' (left: blue rectangle, right: blue dot)|$\Phi_T(\mathbf{z})$, $\Phi_T(\mathbf{z}')$|Teacher mapping at original, perturbed input|
> > > |Student output at z, z' (left: orange rectangle, right: orange dot)|$\Phi_S(\mathbf{z})$, $\Phi_S(\mathbf{z}')$|Student mapping at original, perturbed input|
> > > |Teacher response (right: blue line/arrow)|$\Phi_T(\mathbf{z}') - \Phi_T(\mathbf{z})$|Finite-difference approximation of $J_{\Phi_T}(\mathbf{z}) \mathbf{v}$|
> > > |Student response (right: orange line/arrow)|$\Phi_S(\mathbf{z}') - \Phi_S(\mathbf{z})$|Finite-difference approximation of $J_{\Phi_S}(\mathbf{z}) \mathbf{v}$|
> > > |Response alignment (**Enforcing alignment between the orange line and the blue line on the right.**)|$\| (\Phi_S(\mathbf{z}') - \Phi_S(\mathbf{z})) - (\Phi_T(\mathbf{z}') - \Phi_T(\mathbf{z})) \|_2^2$ | Core GAD loss (finite-difference form)|
> > >
> > > Note: As stated below Eq. 5, the denominator $h$ is absorbed into the loss weighting term, so the figure directly visualizes unscaled numerator vectors.
> > >
> > > We also revise Fig. 3 to explicitly show $L_{\text{GAD}}$ and base distillation loss $L_{\text{base}}$, and include a toy example illustrating geometry recovery for better understanding, available at https://anonymous.4open.science/r/ICML_rebuttal-1490/rebuttal_ICML.pdf.

---

### Official Review · Reviewer_hCwK · 2026-03-11

**Soundness:** 3
**Presentation:** 3
**Significance:** 3
**Originality:** 3
**Overall Recommendation:** 5
**Confidence:** 3

**Summary:**

In the distillation process of text-to-image (T2I) generative models, distillation objectives (such as MSE-based regression loss or reverse KL divergence) primarily focus on pointwise output alignment. This objective encourages the student model to fit the smooth conditional expectation of the multimodal output, inadvertently flattening the input-output landscape and suppressing the local geometric structure inherent in the teacher model.

This paper proposes a framework called Geometry-Aware Distillation (GAD), which abandons single-output matching. Instead, by matching the Jacobian-vector products (JVP) of the teacher and student models with respect to the input noise, GAD forces the student model to restore the local curvature and directional gradients of the teacher model. This successfully recovers sensitivity to initial noise without sacrificing generation quality.

**Compliance With Llm Reviewing Policy:**

Affirmed.

**Final Justification:**

I believe this is a relatively good paper. The problem definition is clear, sufficient experimental validation has been conducted, and during the rebuttal period, additional results were provided to support certain points. I recommend acceptance.

**Key Questions For Authors:**

1. Although it is better than naive second-order computation (which incurs over 200% overhead), the nearly 70% additional time cost remains extremely high for industrial-scale distillation (which often requires thousands of GPU hours). Besides reusing cached conditions, have the authors explored more efficient sampling strategies?

2. When evaluating "General Generation Quality" (Lines 303-316, Table 2), the paper only reports PickScore and CLIP Score on 1000 MS-COCO prompts. These two metrics are biased towards perception and text alignment but cannot fully reflect the fit to the overall data distribution. Could the zero-shot FID on MS-COCO 30K be reported?

3. Should the paper include at least one distillation method explicitly designed to preserve diversity (e.g., the cited Diversity-rewarded CFG distillation) as a comparison baseline?

**Limitations:**

1. Training Overhead: Although the authors strive to reduce computation by reusing cached conditions, the additional forward passes required by finite differences still increase the per-step training time by 36.30% to 66.51%.

2. Theoretical Limitation of Fixed Perturbation Scale: The denoising trajectory of diffusion/flow matching models is a dynamic process (from pure noise to the data manifold), and the mapping curvature varies significantly across different timesteps. Globally fixing the difference perturbation scale, as done in Appendix A, seems overly simplistic.

**Strengths And Weaknesses:**

The experiments effectively demonstrate the method's efficacy: A diagnostic experiment based on Seed Identifiability was designed. After incorporating GAD, this metric improved to 92.40%, closely approaching the performance of the teacher model. The method also maintains General Generation Quality and improves Generation Diversity.

Originality:
Problem Identification: Distillation leads to a loss of sensitivity to initial noise in the model. The paper ingeniously introduces geometric alignment theory into the T2I distillation paradigm using Jacobian-vector product (JVP) matching and Finite Differences, proposing Geometry-Aware Distillation (GAD). The key innovation lies in defining this for the first time as a regularization term for diffusion distillation.

---

> ### Author Rebuttal · Authors · 2026-03-31
>
> > Q1: Efficient sampling strategies beyond caching.
>
> We explore sparse GAD supervision (applying loss to limited timesteps) to further reduce training costs. The table below shows that applying it to the middle 40% of timesteps already outperforms the baseline in semantic alignment and generation quality, while significantly restoring model diversity (Vendi score from 2.59 to 2.71). This demonstrates that GAD delivers strong gains with highly efficient overhead.
>
> |Method|GAD Timesteps|CLIP $\uparrow$|Pickscore $\uparrow$|Vendi $\uparrow$|
> |-|-|-|-|-|
> |Baseline (TDM)|Null|33.43|22.10|2.59|
> |GAD (Sparse)|High noise (>600)|**33.59**|22.24|2.67|
> |GAD (Sparse)| Mid noise (200~600)|33.45|22.13|2.71|
> |GAD (Sparse)| Low noise (<200)|33.32|22.16|2.67|
> |GAD (Sparse)| Random (0.2*Full)|33.10|22.12|2.72|
> |GAD|Full steps|33.52|**22.27**|**2.72**|
>
> > Q2: Zero-shot FID on MS-COCO 30K.
>
> We further evaluate FID scores on the COCO-30K dataset. The table below demonstrates that GAD consistently enhances distributional alignment with the teacher model (lower “FID vs. Teacher”). While GAD leads to a marginal increase in FID when the distilled model already outperforms the teacher (e.g., PixArt), this occurs because GAD shifts the student toward the teacher's geometric distribution, even if the teacher's own FID is slightly higher.
>
> | Model | Method          | FID vs. COCO-30k ↓ | FID vs. Teacher ↓ |
> |-|-|-|-|
> | **SD2**  | Teacher | 16.55            | -                 |
> |          | LADD          | 16.74            | 11.29            |
> |          | **LADD + Ours**| **16.56**        | **7.85**        |
> | **PixArt** | Teacher | 28.49          | -                 |
> |          | TDM           | **25.39**            | 5.80            |
> |          | **TDM + Ours**| 26.32        | **5.14**        |
> | **SANA** | Teacher| 26.51            | -                 |
> |          | SiD           | 28.68            | 8.48            |
> |          | **SiD + Ours**| **27.20**        | **7.36**        |
>
> > Q3: Baselines explicitly preserving diversity.
>
> We note that the mentioned baseline [1] is closed-source and designed specifically for text-to-music tasks. For a fair comparison, we instead evaluate against DDCD [2], a diversity-driven distillation method. DDCD improves diversity by blending teacher and student outputs; however, as shown in the table, it significantly boosts Vendi score at the cost of reduced semantic alignment (CLIP) and generation quality (PickScore), due to the teacher’s limited performance in few-step generation. In contrast, our GAD improves noise sensitivity while further enhancing overall generation quality.
>
>
> | Method | CLIP $\uparrow$ | Pickscore $\uparrow$ | Vendi $\uparrow$ |
> |-|-|-|-|
> | Teacher (PixArt) | 33.41 | 22.29 | 2.73 |
> | Student (TDM) | 33.43 | 22.10 | 2.59 |
> | Student (DDCD [2]) | 33.41 | 22.03 | **2.76** |
> | Student (Ours) | **33.52** | **22.27** | 2.72 |
>
> > L1: Training Overhead.
>
> - **Zero inference cost:** GAD is strictly a training-time modification, incurring zero additional cost during inference.
> - **Superior Efficiency vs. Post-hoc Methods:** Unlike post-hoc optimization methods [3,4], whose costs scale linearly with the number of generated samples, GAD requires only a one-time investment.
> - **Flexible Training:** Sparse GAD (see Q1) reduces training overhead by 60% while maintaining substantial gains, offering a flexible performance-to-cost trade-off.
> - **Fundamental Restoration:** GAD successfully restores the original model's noise sensitivity—the foundation of diversity and controllability—while simultaneously enhancing generation quality during the acceleration of generative models.
>
> > L2: Fixed Perturbation Scale.
>
> To address the reviewer’s concern, we conduct an additional experiment with a timestep-dependent perturbation scale (see table below). We find that gradually increasing the perturbation scale during generation further improves both CLIP score and Vendi diversity. This stems from the intuition that early stages (high noise) amplify even minor perturbations, so a small scale suffices; whereas late stages (low noise) focus on fine detail refinement and are more robust, thus benefiting from a larger perturbation scale.
> | Timestep-Adaptive | CLIP $\uparrow$ | Pickscore $\uparrow$ | Vendi $\uparrow$ |
> |-|-|-|-|
> | Linear (decreasing) | 33.32 | 22.22 | 2.6694 |
> | Linear (increasing) | **33.53** | 22.14 | **2.7202** |
> | Fixed | 33.52 | **22.27** | 2.7187 |
>
> [1] Diversity-Rewarded CFG Distillation. ICLR 2025.
> [2] Distilling Diversity and Control in Diffusion Models. Arxiv 2503.
> [3] SPARKE: Scalable Prompt-Aware Diversity Guidance in Diffusion Models via RKE Score. NeurIPS 2025.
> [4] ReNO: Enhancing One-step Text-to-Image Models through Reward-based Noise Optimization. NeurIPS 2024.

---

> > ### Author Rebuttal · Reviewer_hCwK · 2026-04-02
> >
> > Thank you to the authors for the detailed experiments.
> >
> > I gained the following insights:
> >
> > 1. Applying GAD to the high noisy part of the diffusion timeschedule can significantly restore model diversity.
> >
> > 2. The diffusion timestep is also related to the perturbation scale.
> >
> > I will increase my score.

---

> > > ### Author Response · Authors · 2026-04-06
> > >
> > > Thank you very much for your thoughtful feedback and for confirming that your concerns have been addressed. We sincerely appreciate your careful reading and the insightful observations you summarized regarding timestep selection and perturbation scale. We will incorporate the corresponding experiments and analysis into the revised version. We are also grateful for your support and for considering an increased score.

---

### Official Review · Reviewer_f9SB · 2026-03-13

**Soundness:** 4
**Presentation:** 4
**Significance:** 3
**Originality:** 4
**Overall Recommendation:** 5
**Confidence:** 3

**Summary:**

This paper firstly presents a new finding that distilled text-to-image diffusion models lose the noise-to-image mapping in their teacher models. Based on this finding, it proposes to align not only the score but also the Jacobian of the score in diffusion models. At a cost of an extra forwarding, this method successfully restores the noise sensitivity of teacher text-to-image diffusion models in the students.

**Compliance With Llm Reviewing Policy:**

Affirmed.

**Final Justification:**

I keep my positive score.

**Key Questions For Authors:**

N/A

**Limitations:**

yes

**Strengths And Weaknesses:**

Strengths:

[1] The finding that distilled text-to-image diffusion models lose the noise-to-image mapping in their teacher models is valid and interesting, matching the insight of smoothing effect of distillation.

[2] This paper targets a significant problem: controllability of distilled (few-step) text-to-image diffusion models.

[3] The proposed method is both intuitive and empirically effective, with flexibility to be adapted to different branches of distillation methods.

[4] Experiments are sufficient.

Weaknesses: I don’t see significant weaknesses of this paper. Good paper.

---

> ### Author Rebuttal · Authors · 2026-03-31
>
> We sincerely thank the reviewer for the positive and encouraging assessment of our work.
>
> We are glad that the reviewer finds our key observation (the loss of noise-to-image mapping in distilled models) both valid and insightful, and agrees with its connection to the smoothing effect of distillation. We also appreciate the recognition of the importance of restoring controllability in few-step text-to-image models.
>
> We are particularly encouraged that the reviewer finds our proposed method intuitive, flexible, and broadly applicable across different distillation paradigms.
>
> Finally, we thank the reviewer for acknowledging the overall soundness, empirical validation, and potential impact of our work. We will incorporate additional clarifications and minor improvements based on all reviewers’ feedback to further strengthen the paper.

---

> > ### Author Rebuttal · Reviewer_f9SB · 2026-04-03
> >
> > NA

---

> > > ### Author Response · Authors · 2026-04-06
> > >
> > > Thank you very much for your positive feedback. We sincerely appreciate your careful evaluation and support.

---

### Official Review · Reviewer_BEpg · 2026-03-13

**Soundness:** 3
**Presentation:** 2
**Significance:** 3
**Originality:** 3
**Overall Recommendation:** 5
**Confidence:** 4

**Summary:**

This paper identifies and addresses the problem of degraded sensitivity to initial noise in distilled text-to-image diffusion models. The authors observe that standard distillation objectives (e.g., MSE, KL divergence) focus on pointwise output alignment, which inadvertently smooths the input-output landscape and suppresses the local geometric structure—particularly the differential response to noise perturbations—that the teacher model possesses. This leads to reduced diversity, weakened controllability via noise manipulation, and failure of downstream noise-based control techniques. To address this issue, they propose Geometry-Aware Distillation (GAD), a plug-and-play regularization term that aligns Jacobian-vector products (JVPs) between teacher and student. To avoid the prohibitive cost of full Jacobian computation, they approximate JVPs via finite differences, yielding a practical loss that compares "response vectors" (output differences under paired noise perturbations). GAD is demonstrated across three architectures (SD2/UNet, PixArt-α/DiT, SANA/Flow-DiT) and three distillation paradigms (output matching, distribution matching, score identity distillation), with evaluations on seed identifiability, layout control, generation diversity, and zero-shot controllability transfer.

**Compliance With Llm Reviewing Policy:**

Affirmed.

**Final Justification:**

I found that the definition of the "Score-based" version of the GAD loss presented in the submitted manuscript is a scalar loss; the authors' explanation and writing regarding this part need to be strengthened to clarify the actual algorithm. The explanation of why GAD can improve model performance is interesting and seems somewhat analogous to TDM in that it enhances the model's ability to retain the teacher trajectory. Overall, I still believe this paper has the potential to become a foundational component for fast models and is worthy of acceptance.

**Key Questions For Authors:**

- The experiments include both one-step (LADD on SD2) and few-step (TDM) settings. Do you observe systematic differences in how much sensitivity GAD can recover as a function of the number of student steps?  Intuitively, one-step distillation potentially makes geometric alignment harder.
- See Weaknesses

**Limitations:**

- While the finite-difference approximation is practical, the choice of perturbation scale appears to be quite sensitive and architecture-dependent (Table 7 shows h=0.0001 for SD2.1 but h =0 .01 for PixArt-α and SANA — a 100× difference).

**Strengths And Weaknesses:**

Strengths

- Overall, the authors explore the concept of noise sensitivity degradation in distilled models. The diagnostic experiment in Section 3 (Figure 2) is effective: showing that low pointwise MSE coexists with high JVP error is a clean and convincing way to motivate the approach.

- The core idea is neat and interesting. The finite-difference approximation of the JVP alignment is both computationally tractable and broadly applicable.

- The experiments are extensive, validating the proposed method across different distillation algorithms and backbones.

Weaknesses

- The writing needs improvement. For example, I am confused about how GDA is applied to TDM. Based on the paper, it seems like Score-based Alignment should be used, but upon reviewing the code, I found that it actually applies the output matching version of GDA based on a surrogate loss derived from TDM.

- The mechanism by which Score-based Alignment regularizes the generator is unclear. If my understanding is correct, Score-based Alignment operates on the fake score, which is trained to follow the student's distribution, so this regularization cannot directly act on the student. The authors should at least provide some intuitive explanation in the paper for why it works.

- The training overhead is non-negligible. Table 6 shows that GAD increases wall-clock time by 36–67% and memory by 4–47% depending on the setting. Thus, a question is: how does GDA perform when the total training cost is controlled to be comparable?

---

> ### Author Rebuttal · Authors · 2026-03-31
>
> > W1: Details about how GAD is applied to TDM.
>
> We appreciate the reviewer's careful look at our code and the opportunity to clarify the implementation of GAD within the TDM framework.
>
> - **Clarification on "Score-based" vs. "Output Matching"**: "Output matching" in our text describes the computational procedure of averaging per-sample losses. In TDM, the real/fake scores are the processed "outputs" of the score-matching networks.
> - **Preservation of Base Distillation:** GAD acts as a plug-and-play regularizer that does not modify the underlying alignment of the base distillation method. For TDM, we strictly follow its original distribution matching paradigm, which aligns the real and fake score fields.
> - **From Pointwise to Relational Alignment**: Beyond standard pointwise alignment, GAD introduces a small perturbation (line 1424 of the code). We enforce that the **variation** in the real score field matches the **variation** in the fake score field (lines 1415–1478 of the code). This adds pairwise relative relationship alignment without modifying TDM’s underlying score-matching logic.
>
> > W2: How does score-based alignment regularize the generator?
>
> To clarify, while both score-based alignment and **GAD loss** achieve alignment by comparing real and fake scores, the loss is exclusively used to optimize the **student generator $\Phi_S$**, not the fake model. This ensures that the **GAD regularization** effectively guides the student model's learning through a **differentiable gradient path**.
>
> - **Differentiable Gradient Path**: In score-based paradigms, generator $\Phi_S$ is updated by minimizing a divergence dependent on score estimators. As samples $x_t$ are outputs of $\Phi_S$, the GAD loss (Eq. 9) computed on $x_t$ is fully differentiable with respect to student parameters $\theta$:
> $$
> \mathcal{L}_{\text{GAD}}^{\text{score}} = \mathbb{E}_{\mathbf{x}_t, \mathbf{v}, t, c} \left[ \left\| \Delta \epsilon_S(\mathbf{x}_t, \mathbf{v}) - \text{sg}\left( \Delta \epsilon_T(\mathbf{x}_t, \mathbf{v}) \right) \right\|_2^2 \right] =
>  \mathbb{E}_{\mathbf{x}_t, \mathbf{v}, t, c} \left[ \left\| \Delta \epsilon_S(\Phi_{S}(\mathbf{z};\theta ), \mathbf{v}) - \text{sg}\left( \Delta \epsilon_T(\Phi_{S}(\mathbf{z};\theta ), \mathbf{v}) \right) \right\|_2^2 \right]
> $$
>
> In practice, the student and the fake score estimator evolve together. Intuitively, GAD forces the student to produce a distribution whose score field $\epsilon_S$ replicates the teacher’s local curvature and divergence. If the student manifold is "flat" (over-smoothed), the discrepancy in $\Delta\epsilon$ provides a corrective gradient that pushes the student to restore the geometric complexity inherited from the teacher.
>
> > W3: Performance under comparable training costs.
>
> We scale the baseline TDM’s training time and peak memory (via batch size) to match GAD. We also report GAD (Sparse), an efficient variant per our response to Reviewer hCwK (Q1). The table below shows that GAD consistently outperforms TDM under equal budgets. This suggests that even with increased training resources, TDM remains constrained by the over-smoothing effect inherent in pointwise distillation. In contrast, our geometric alignment enables the model to achieve superior peak performance upon convergence.
>
> |Method|Training Time|Peak Memory|CLIP $\uparrow$|Pickscore $\uparrow$|Vendi $\uparrow$|
> |-|-|-|-|-|-|
> |TDM (Baseline)|1.0x|1.0x|33.43|22.10|2.59|
> |TDM (Comparable)|1.5x|1.2x|33.35|22.19|2.68|
> |GAD (Sparse)|1.1x|1.2x|33.45|22.13|2.71|
> |GAD (Ours)|1.5x|1.2x|**33.52**|**22.27**|**2.72**|
>
> > Q1: Performance across different student steps.
>
> We conduct an additional ablation study on PixArt-$\alpha$ across 1-, 2-, and 4-step student models. The table below shows that while 1-step models are more challenging to align, they also suffer from the most severe sensitivity collapse. GAD successfully mitigates this across all regimes, consistently improving diversity (Vendi Score).
>
> |Method|Steps|CLIP $\uparrow$|Pickscore $\uparrow$|Vendi $\uparrow$|
> |-|-|-|-|-|
> |Teacher|20|33.41|22.29|2.73|
> |TDM|4|33.43|22.10|2.59|
> |TDM + Ours|4|**33.52**|**22.27**|**2.72** (+4.70%)|
> |TDM|2|31.04|21.54|2.61|
> |TDM + Ours|2|**32.23**|**21.78**|**2.67** (+2.44%)|
> |TDM|1|29.61|20.47|2.05|
> |TDM + Ours|1|**29.82**|**20.52**|**2.09** (+1.70%)|
>
> > L1: Architecture-dependent perturbation scale
>
> We observe that the optimal perturbation scale $h$ correlates with model sensitivity: sensitive models require smaller $h$, while robust ones need larger. To investigate this, we compare the response of SD2.1 and PixArt to an identical input perturbation. SD2.1 shows 5.5% output change under scale 0.01 perturbation versus PixArt’s 1.8%, indicating higher sensitivity and thus needing larger $h$ for alignment. Furthermore, as illustrated in Fig.7 of our original manuscript, GAD exhibits a wide robustness interval for the perturbation scale, remaining effective across a range from $1e^{-1}$ to $1e^{-4}$.

---

> > ### Author Rebuttal · Reviewer_BEpg · 2026-03-31
> >
> > I appreciate this neat idea and believe that, benefiting from its simplicity, it has the potential to become an essential component for future distilled models.
> >
> > However, I still do not understand the authors' formulation of the GAD loss for score-based alignment.
> >
> > From the equation, it appears that we need to backpropagate gradients through the fake score; yet in the code implementation, the GAD loss does not involve gradient backpropagation through the fake score. The authors should provide more intuitive loss formulations and clearly label which model each term corresponds to.
> >
> > Additionally, while GAD is designed to improve diversity, it also consistently enhances generation quality in practice. How do the authors explain this phenomenon?
> >
> > If the authors provide a convincing response, I will raise my rating to 5 to support this paper.

---

> > > ### Author Response · Authors · 2026-04-06
> > >
> > > > Q1: Score-based GAD loss implementation.
> > >
> > > We thank the reviewer for pointing out the potential confusion in our GAD formulation.
> > >
> > > - **For Formulation.** We clarify that the original formulation of GAD loss in score-based alignment (Eq.9) defines the **gradient** of the GAD loss w.r.t the student’s parameters $\theta$, rather than a **scalar loss**. We will revise Eq.9 to $$\nabla_{\theta}\mathcal{L}_{\text{GAD}}^{\text{score}} = \mathbb{E}_{\mathbf{x}_t, \mathbf{v}, t, c} [ \Delta \epsilon_{\text{fake}}(\mathbf{x}_t, \mathbf{v}) - \Delta \epsilon_{\text{real}}(\mathbf{x}_t, \mathbf{v}) ]\frac{\partial \mathbf{x}_t}{\partial \theta},$$ where $\epsilon_{\text{fake}}(\cdot)$ is the fake score estimator, $\epsilon_{\text{real}}(\cdot)$ is the teacher, and $\mathbf{x}_t$ is the sample generated by the student $\Phi_S$, which depends on parameter $\theta$. $\mathbf{v}$ is the random perturbation direction, $t$ is the timestep and $c$ is the text condition.
> > >
> > > - **For Code Implementation.** Since score-based alignment defines a target gradient rather than a scalar loss [1,2], it does not directly specify gradients in code but instead constructs a **surrogate loss** designed to yield the exact target gradient when differentiated. Specifically, for the target **gradient** $\mathbf{g} = \epsilon_{\text{fake}} - \epsilon_{\text{real}}$, we construct a surrogate MSE loss: $\mathcal{L}_{\text{surrogate}} = \frac{1}{2}\|\mathbf{x} - \text{sg}(\mathbf{x} - \mathbf{g})\|_2^2,$ where $\mathbf{x}$ (shorthand for $\mathbf{x}_t$) is the sample generated by student $\Phi_S(\theta)$. Let $\mathbf{y} = \text{sg}(\mathbf{x} - \mathbf{g})$. The gradient w.r.t. $\mathbf{x}$ is:
> > > $$
> > > \nabla_{\mathbf{x}} \mathcal{L}_{\text{surrogate}}
> > > = \frac{\partial \mathcal{L}}{\partial (\mathbf{x}-\mathbf{y})}
> > > \cdot \frac{\partial (\mathbf{x}-\mathbf{y})}{\partial \mathbf{x}}
> > > = \mathbf{x} - \mathbf{y}=\mathbf{g},
> > > $$
> > > since $\frac{\partial \mathbf{y}}{\partial \mathbf{x}} = 0$ due to the stop-gradient operator. Thus, optimizing $\mathcal{L}_{\text{surrogate}}$ induces the desired gradient update on $\mathbf{x}$.
> > >
> > > - **GAD's Gradient Path**: GAD follows this surrogate formulation, with $\mathbf{g}$ replaced by aligned **response (score changes**).
> > > 1. Teacher/fake response $\Delta \epsilon_{\text{real}/\text{fake}} = \epsilon_{\text{real}/\text{fake}}(\mathbf{x} + h\mathbf{v}) - \epsilon_{\text{real}/\text{fake}}(\mathbf{x})$ are computed in `no_grad()` (Code: 1427–1439, 1448-1450).
> > > 2. The student response $\Delta \mathbf{x} = \Phi_S(\mathbf{x} + h\mathbf{v};\theta) - \Phi_S(\mathbf{x};\theta)$ is computed with gradients enabled (Code: 1451). The GAD surrogate loss is then
> > > $$
> > > \mathcal{L}_{\text{GAD}} = \|\Delta \mathbf{x} - \text{sg}(\Delta \mathbf{x} -( \Delta \epsilon_{\text{fake}} - \Delta \epsilon_{\text{real}}))\|_2^2
> > > $$
> > > (Code: 1472). Since $\Delta \mathbf{x}$ depends on $\theta$, **gradients flow only through the first term $\Delta \mathbf{x}$ (the student path), while the fake score estimator $\Delta \epsilon_{\text{fake}}$ is blocked by sg().**
> > >
> > > |Notation|Explanation|Code|
> > > |-|-|-|
> > > |$\Delta \mathbf{x}$|finite-difference change of student output|Line 1451|
> > > |$\Delta \epsilon_{\text{real}}$|finite-difference change of real score|Line 1428-1435, 1448-1449
> > > |$\Delta \epsilon_{\text{fake}}$|finite-difference change of fake score|Line 1436-1439, 1450|
> > > |$\Delta \mathbf{x} - \text{sg}(\cdot)$|Alignment term in surrogate loss|Line 1472|
> > >
> > > > Q2: Generation quality improvement.
> > >
> > > - **Reason.** We attribute the quality improvement to **better generalization** of GAD during inference. Standard pointwise distillation treats each input independently, ignoring the local structure of the generative manifold, which can lead to geometric distortion (overfitting to isolated samples and oscillating between them). In contrast, GAD enforces local neighborhood consistency via Jacobian-vector alignment, resulting in a more faithful approximation of the manifold. Consequently, when encountering unseen inputs during inference, the model produces denoising trajectories more consistent with the teacher, thereby improving generation quality.
> > > - **Evidence.** We further measure the average cumulative trajectory deviation from the teacher’s denoising path (discrepancy between the denoised latents) on 200 COCO prompts (unseen during training) using PixArt. As shown below, GAD consistently reduces the deviation from the teacher at every stage, achieving a 13% lower final error. This confirms that GAD better preserves teacher-consistent denoising dynamics during inference, which explains the observed quality improvement.
> > >
> > > |Method|Stage 1 Drift (t=0.75) $\downarrow$|Stage 2 Drift (t=0.5) $\downarrow$|Stage 3 Drift (t=0.25)$\downarrow$|Stage 4 Drift (Final)$\downarrow$|
> > > |-|-|-|-|-|
> > > |Baseline|0.016|0.216|0.433|0.491|
> > > |+ GAD|**0.014**|**0.184**|**0.373**|**0.427**|
> > >
> > > [1] DMD (Distribution Matching Distillation). CVPR 2024.
> > > [2] TDM (Trajectory Distribution Matching). Arxiv 2503.

---

### Decision · Program_Chairs · 2026-04-30

**Decision:**

Accept (regular)

**Comment:**

This paper identifies a previously overlooked but practically important property lost during diffusion model distillation  and proposes Geometry-Aware Distillation (GAD), a plug-and-play regularization framework that restores this sensitivity by aligning Jacobian-vector products between teacher and student models via efficient finite-difference approximation. The paper is well-motivated, clearly presented, and makes a meaningful contribution to the growing field of few-step text-to-image generation.

Three out of four reviewers recommend acceptance, with two assigning the highest score of 5 and a third upgrading to 5 following a thorough rebuttal. The reviewers broadly agree on the validity of the core observation, the mathematical soundness of the proposed approach, and the breadth and quality of the experimental evaluation spanning multiple architectures. The single dissenting reviewer (weak reject, score 3) raised concerns, but notably declared a low confidence level of 2 out of 5, and these concerns were not shared by any of the other reviewers.

Overall, the contribution is technically sound, practically significant, and sufficiently validated, and the paper is recommended for acceptance.